# Neural Thickets:
## Diverse Task Experts Are Dense Around Pretrained Weights

**Yulu Gan** [1]   **Phillip Isola** [1]

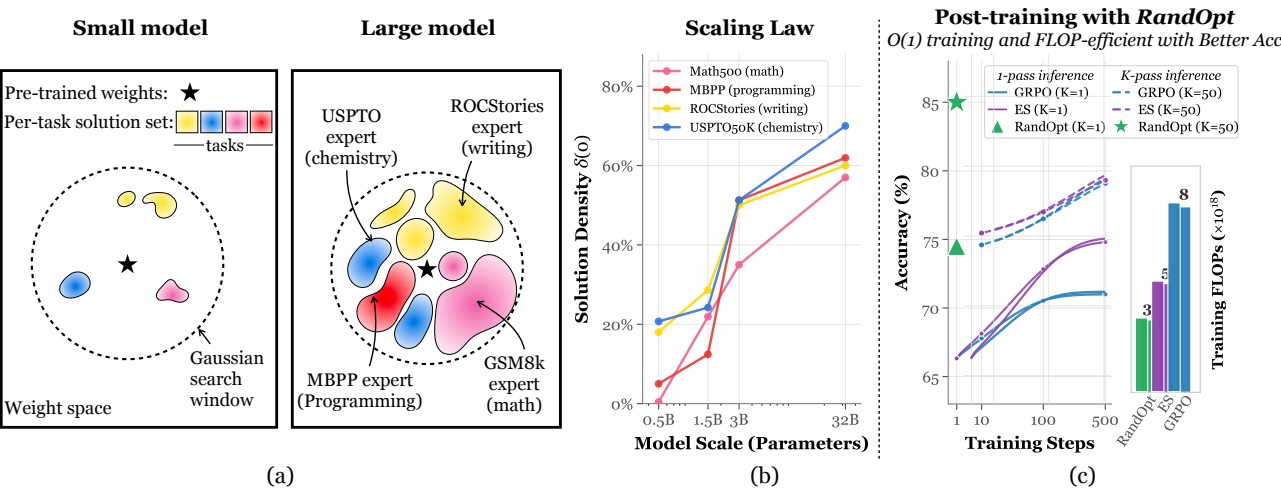

*Figure 1.* (a) Schematic of the main effects we observe (see Fig 2 for a version with real data). Left: Small models live in a *needle in a haystack* regime, where good solutions to downstream tasks occupy a tiny fraction of the surrounding weights. In this regime, it is important to have a smart search algorithm, such as gradient descent or other forms of iterative optimization. Right: Large models are surrounded by a veritable *thicket* of task-specific solutions. In this regime, random sampling is sufficient to quickly land on promising adaptations, which can then be ensembled to yield strong behavior, an approach we call *RandOpt*. (b) Solution density – i.e. density of task-improving weights in a Gaussian neighborhood of the pretrained weights – scales with model size. (c) RandOpt is $\mathcal{O}(1)$ in training steps, FLOP-efficient, and competitive in converged accuracy with GRPO and ES. Results are shown on the Countdown task with `Olmo-3-7B-Instruct`; RandOpt uses 5000 random weight guesses and ensembles the top $K$; K-pass baselines use Test-time Majority Vote (TT-MV). More results are shown in Fig. 6 and Table 4.

## Abstract

Pretraining produces a learned parameter vector that is typically treated as a starting point for further iterative adaptation. In this work, we instead view the outcome of pretraining as a distribution over parameter vectors, whose support already contains task-specific experts. We show that in small models such expert solutions occupy a negligible fraction of the volume of this distribution, making their discovery reliant on structured optimization methods such as gradient descent. In contrast, in large, well-pretrained models the density of task-experts increases dramatically, so that diverse, task-improving specialists populate a substantial fraction of the neighborhood around the pretrained weights. Motivated by this perspective, we explore a simple, fully parallel post-training method that samples $N$ parameter perturbations at random, selects the top $K$, and ensembles predictions via majority vote. Despite its simplicity, this approach is competitive with standard post-training methods such as PPO, GRPO, and ES for contemporary large-scale models.

Project page: https://thickets.mit.edu

Code: https://github.com/sunrainyg/RandOpt

[1]MIT CSAIL. Correspondence to: Yulu Gan <yulu_gan@mit.edu>, Phillip Isola <phillipi@mit.edu>.

*Proceedings of the $43^{rd}$ International Conference on Machine Learning*, Seoul, South Korea. PMLR 306, 2026. Copyright 2026 by the author(s).

# 1. Introduction

*"[Random guessing] cannot be viewed as a reasonable learning algorithm..."*
— Schmidhuber, Hochreiter, Bengio, 2001

One of the first algorithms we learn in elementary school is "guess and check." In its simplest form the procedure is almost trivial: given an equation with unknowns, guess values of the unknowns and check whether they satisfy the equation. Guess again, completely at random, until it works. The same approach can be applied to machine learning, but it has long been assumed to be hopeless, as the quote above implies (Schmidhuber et al., 2001). Consider, for example, the chance of randomly guessing, from scratch, a billion-dimensional parameter vector that behaves like ChatGPT. The probability must be astronomically small.

This paper finds that after pretraining, the story changes. With a reasonable number of random guesses, one can sample parameter perturbations that substantially improve pretrained large language models (LLMs) across many tasks. How is this possible? For random guessing to work, good solutions must be dense under the distribution being sampled from. Schmidhuber, Hochreiter, and Bengio made precisely this point in their 2001 paper: random guessing *did* solve some benchmark problems of that era. However, they interpreted this as a failure of the benchmarks to assess difficult skills. We instead show that the same phenomenon occurs in a contemporary setting of practical interest: post-training LLMs on tasks such as reasoning, programming, and more.

How does the loss landscape change after pretraining, in order that random guessing begins to work? We study two effects. First we measure the *density* of task-improving solutions in a Gaussian neighborhood around the pretrained weights. We find that this density increases with pretraining scale. Untrained models have a tiny density of solutions near their initial weights; they live in a *needle in a haystack* regime, where finding the solution requires structured multi-step search, such as gradient descent. Conversely, large pretrained models transition into a regime of high density, replete with task-experts near the pretrained weights. We term this the *thicket regime*.

Next, we study solution *diversity* in the Gaussian neighborhood. It turns out that the different sampled parameter vectors are not uniform improvements. They are specialists rather than generalists, where the perturbations that most improve performance on one task hurt performance on other tasks. We visualize these two effects in Figure 1.

Motivated by these findings, we explore a fully parallel post-training algorithm that exploits the density and diversity of pretrained neighborhoods. The method is a form of random guessing (which works due to the neighborhood density) followed by ensembling (which exploits the neighborhood

diversity). Given an initial weight vector, $N$ perturbations of that vector are created. Each perturbation is evaluated on the post-training data. The top $K$ perturbations are selected and their predictions are ensembled via majority voting. We call this algorithm RandOpt.

RandOpt achieves accuracy competitive with PPO, GRPO, and ES under the same post-training flops. On the wall-clock it trains in $\mathcal{O}(1)$ time compared to $\mathcal{O}(T)$ for the baselines that require $T$ sequential update steps. Its inference-time cost is $K$ times higher due to ensembling. For some tasks, $K = 1$ already achieves decent results. For other tasks, only larger $K$ is competitive with the baselines, but in Section 6, we show a proof of concept that this cost can be reduced via distillation. However, our goal is not to promote RandOpt as superior to alternative methods. Rather, we use it as a probe: its success suggests that *post-training becomes easy once you have a strong pretrained representation* – i.e., once you enter the thicket regime. In that regime, it doesn't matter much which method you use – gradient-based search, evolutionary algorithms, and brute-force parallel selection all will do.

**Main Findings**

1. In large models, the neighborhood around pretrained weights is dense with task-improving solutions (Fig 2).

2. The density exhibits a scaling law, with higher density for larger, more performant models (Fig 3).

3. The local neighborhood is also diverse: individual perturbations tend to improve performance on particular tasks while degrading others (Fig 4). Diversity also scales with model size (Fig 3).

4. For current models, the density is high enough that random guessing is effective for post-training (Fig 6).

5. Ensembling over multiple guessed solutions further improves performance, often substantially (Fig 10).

**Conflict of Interest Disclosure.** The authors declare no financial conflicts of interest.

# 2. Structure of the Multi-task Loss Landscape Around Pretrained Weights

In this section, we measure the density and diversity of task-improving solutions in the vicinity of pretrained weights, across models of different scale and a variety of tasks.

## 2.1. Solution Density: What Proportion of Local Perturbations Improve Task Performance?

Figure 2 visualizes the performance of Gaussian-perturbed models across scales from 0.5B to 32B parameters, and on

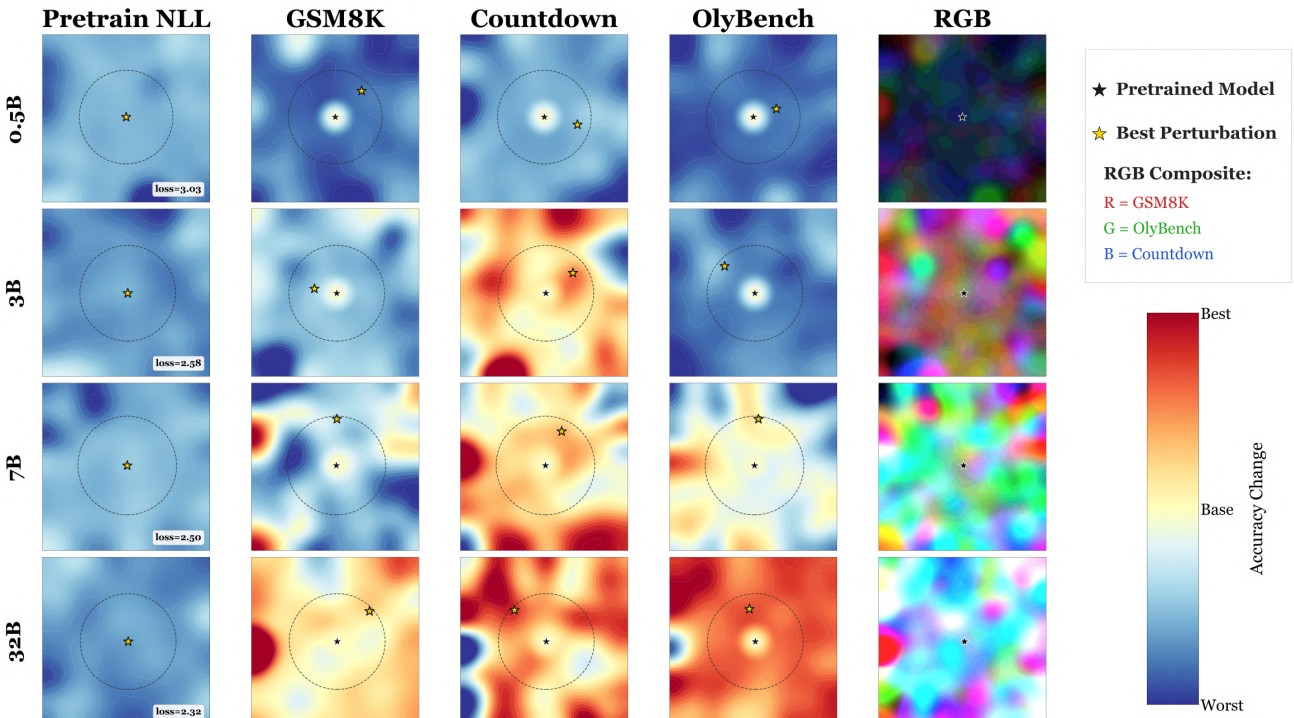

*Figure 2.* Accuracy landscapes in weight space across model scales and reasoning tasks. We perturb the pretrained Qwen2.5 models (from 0.5B to 32B) with 1000 random weight perturbations and project the perturbed models into 2D using random projection. Colors show relative accuracy change (acc − base)/base × 100 (blue: degraded, white: equivalent, red: improved.) Dashed circles indicate the mean perturbation distance and stars mark the best-performing perturbations. Larger models have warmer landscapes, with more high-performing neighborhoods. The last column shows an RGB visualization where GSM8K, Olympiad and Countdown accuracies are mapped to R,G,B channels; richer colors indicate more task experts.

three reasoning tasks. A topographic shift is evident: small models reside on local maxima of the accuracy landscape, whereas larger models inhabit an accuracy "valley," with many peaks of higher accuracy nearby (red regions). This indicates that scaling may fundamentally reshape the loss landscape. Given this change, what is the probability of finding good solutions? We define the solution density as:

**Definition 2.1** (Solution Density). Let $s : \mathbb{R}^d \to \mathbb{R}$ be the performance evaluation metric for model parameters $\boldsymbol{\theta} \in \mathbb{R}^d$. We define the *Solution Density* $\delta(m)$ as the probability that a random perturbation $\boldsymbol{\epsilon}$ improves the base model's score by a margin $m$:

$$\delta(m) = \mathbb{P}_{\boldsymbol{\epsilon} \sim \mathcal{N}(\mathbf{0}, \sigma^2 \mathbf{I})} \left[ s(\boldsymbol{\theta} + \boldsymbol{\epsilon}) \geq s(\boldsymbol{\theta}) + m \right] \quad (1)$$

where $\sigma$ scales the local Gaussian neighborhood (in the experiments in this section, we use $\sigma = 0.005$). Intuitively, $\delta(m)$ measures the "hit rate" of random guessing: it quantifies the proportion of the explored parameter space that yields a performance gain of at least $m$.

In Figure 3, we measure $\delta(m)$, for several values of $m$, as a function of model scale. Solution density increases monotonically with model size, and this trend holds over multiple values of $m$ (e.g., the density of perturbations that increase

performance by +5% accuracy increases monotonically as a function of model scale). These results indicate that, for large-scale models, the pretrained weights reside within a dense basin populated by abundant high-quality solutions, whose density scales with model size.

## 2.2. Solution Diversity: Are the Sampled Perturbations Specialists or Generalists?

Do all perturbations help (or hurt) in the same way? We compare two hypotheses:

1. Hypothesis 1 (generalists): The pretrained weights are in fact a poor model for the suite of downstream tasks we are testing; there is an all-around better model in the vicinity of these weights.
2. Hypothesis 2 (specialists): The pretrained weights are a "jack of all trades, master of none"; perturbations can improve a given task because they are specialists for that task, improving its performance while hurting performance on other tasks.

We test these hypotheses using the following measure of specialization:

**Definition 2.2** (Spectral Discordance). Let $\mathbf{P} \in [0, 1]^{N \times M}$

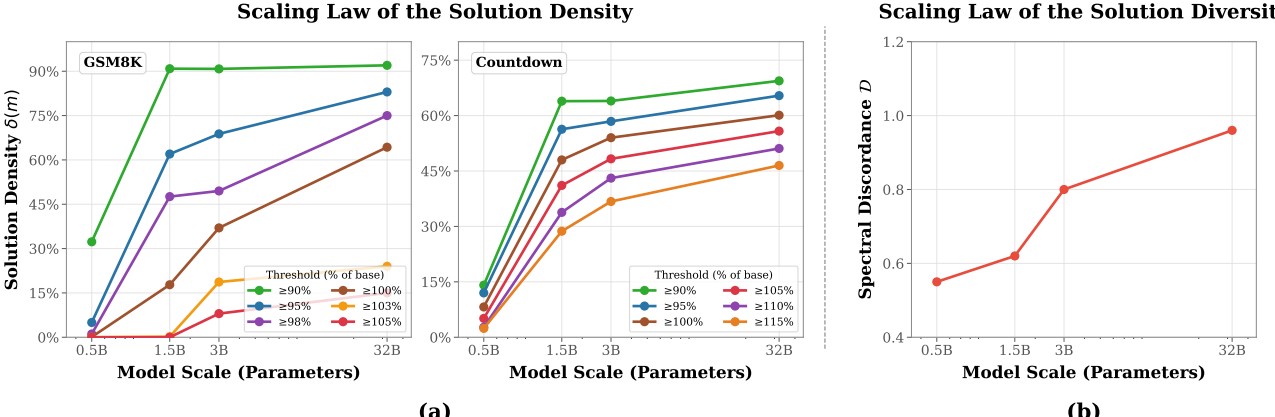

**(a)**

*Figure 3.* Scaling laws of solution density and diversity (using *Qwen-2.5* instruction tuned models). (a) Solution density increases with model scale, showing that larger models have a higher fraction of good solutions. (b) Spectral discordance across model scales, measuring solution diversity. Together, these results demonstrate that larger models have both denser and more diverse solution landscapes in the neighborhood around their pretrained weights.

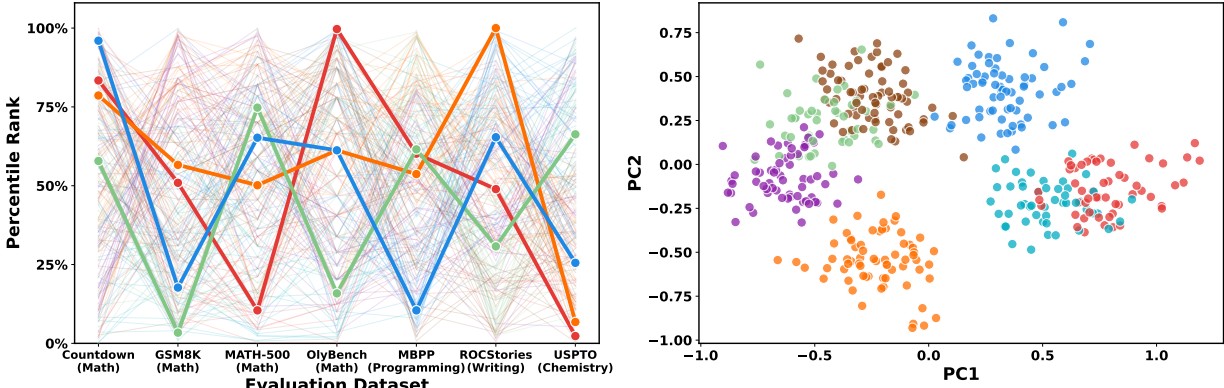

*Figure 4.* Performance spectra and clustering of random seeds. Sampled vectors possess diverse areas of expertise, with individual seeds specializing in specific tasks. (Left) Performance of 100 random seeds across seven evaluation datasets. Each line represents a specific seed, with four lines highlighted as examples. (Right) PCA visualization of these performance vectors, where seeds with similar behavior cluster together into different groups.

denote the percentile-rank matrix over $N$ seeds and $M$ tasks, and let $\mathbf{C} \in \mathbb{R}^{M \times M}$ be the Pearson correlation matrix of its columns. We define the *Spectral Discordance* as:

$$\mathcal{D} = 1 - \frac{1}{M(M-1)} \sum_{j \neq k} \mathbf{C}_{jk} \quad (2)$$

where $\mathbf{C}_{jk}$ denotes the correlation between tasks $j$ and $k$. A value of $\mathcal{D} \rightarrow 1$ implies orthogonal task rankings (specialists), while $\mathcal{D} \rightarrow 0$ implies parallel rankings (generalists).

*Remark.* Theoretically, $\mathcal{D}$ is bounded in the interval $[0, \frac{M}{M-1}]$. The upper bound corresponds to the limit of anti-correlation. We provide the derivation in Appendix L.

We evaluate 500 random weight perturbations across seven tasks in four domains: mathematical reasoning (Countdown, GSM8K, MATH-500, OlympiadBench), code generation (MBPP), creative writing (ROCStories), and chemistry

(USPTO). For each perturbation, we compute its percentile rank relative to the population. We then quantify diversity using the *Spectral Discordance* $\mathcal{D}$ over all the perturbations. Figure 3(b) shows the result over models of different size in the Qwen2.5 family: $\mathcal{D}$ increases monotonically with model size, indicating that the solutions surrounding larger models become increasingly disjoint in their capabilities. This supports Hypothesis 2: the sampled solutions are specialists.

To visually unpack the structure of this discordance, Figures 4 and 2 present further analyses:

(1) Performance Spectra (Figure 4 Left): We plot the percentile ranks of individual perturbations across tasks. The resulting lines are "spiky" rather than flat. This visually demonstrates specialization and diversity among the sampled perturbations.

(2) PCA Projection (Figure 4 Right): We project the 7-

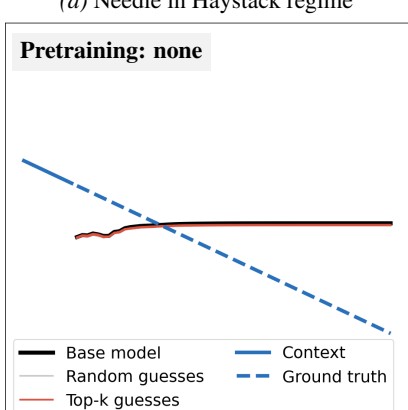
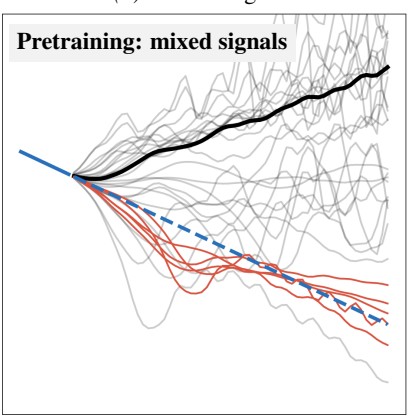
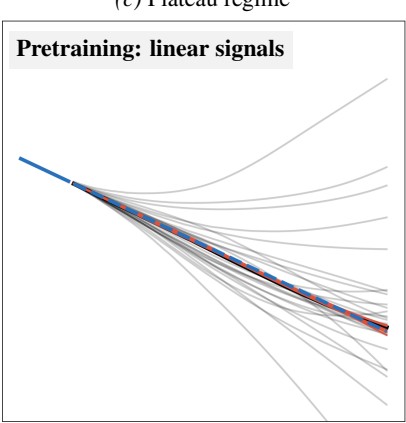

*Figure 5.* Pretraining a model of 1D signals, then probing the local neighborhood around the pretrained weights by random guessing $N = 1000$ Gaussian perturbations. The plot shows the autoregressive predictions of a particular linear function (dashed blue line), given an observed context (solid blue line). Gray lines: random $f_\theta$'s; Red lines: top-K $f_\theta$'s. The figure shows three regimes: (a) No pretraining leads to needle-in-the-haystack search, (b) pretraining on several signal types leads to a thicket, (c) pretraining on just linear functions achieves nearly perfect predictions at pretraining time, hence post-training is at ceiling.

dimensional performance vectors into 2D and apply K-means clustering. The emergence of distinct clusters confirms that there are multiple kinds of experts within the distribution. Perturbations within a cluster share specific strengths (e.g., excelling at math but failing at chemistry), while perturbations in different clusters offer complementary capabilities.

(3) The level of diversity is also visible in Figure 2. Under the column labeled "RGB", we plot three different task-accuracy landscapes in the R, G, and B channels respectively. The fact that there is a mottled, multi-colored appearance indicates that the landscapes are largely uncorrelated (we would expect to see shades of gray if all tasks behaved the same).

These findings reveal that the local weight space is populated by diverse specialists. This raises a key question: can we exploit this landscape by simply sampling these complementary experts and aggregating their strengths? We will return to this question in Section 4.

## 3. A Minimal Setting In Which Thickets Emerge: Autoregressive Modeling of 1D Signals

What leads to thickets emerging? To elucidate the cause, we replicate the phenomenon in a minimal setting.

We define a training distribution that is a mixture of several random function types (sinusoidal, linear, harmonic, sigmoidal, and sawtooth and square waves), each of which maps $\mathbb{R} \to \mathbb{R}$ and takes on random settings for the function parameters (e.g., phase and amplitude for sinusoids, slope and intercept for lines). We pretrain a next-value

predictor on these functions, using a multilayer perceptron $f_\theta : \mathbf{y}_{\text{CTX}} \to y_{\text{NEXT}}$, where $\mathbf{y}_{\text{CTX}}$ is a preceding context window and $y_{\text{NEXT}}$ is the next value of the target function. This model can generate predictions by autoregressive rollout given an initial observed context. We probe this model with a simple linear test signal. Is this signal well-modeled by sampled perturbations near the pretrained weights?

We sample $N = 1000$ random Gaussian parameter perturbations from $\epsilon \sim \mathcal{N}(0, \sigma = 0.002)$. In Figure 5, we show autoregressive rollouts of the base and perturbed models given the test context. The blue line is the test function, with solid blue as the observed context and dashed blue as the ground truth continuation. The gray lines are predictions given by different random perturbations. The base, unperturbed model's predictions are shown in black. The top 5 perturbations that best fit the blue line are shown in red.

We compare three pretraining settings: 1) no pretraining (Xavier initialization, Glorot & Bengio (2010)), 2) pretraining on all signal types, 3) pretraining on linear signals. These three settings lead to three different regimes: 1) the needle in the haystack regime, where small perturbations of the weights have negligible effect on the functional shape; good solutions are far away from this initialization, 2) the thicket regime, where different perturbations search over many possible continuations following the kinds of functions seen during pretraining, and 3) a "plateau" regime, where the pretrained weights are already a minimizer of the test task, and random guessing can provide no further benefit.

This experiment demonstrates that the phenomena we are observing are not exclusive to LLMs, and show up in simpler models as well. What appears to be critical is that the base

---

**Algorithm 1** RandOpt (PyTorch-style). **N**: population size, **K**: ensemble size. **sigmas**: noise scales, **theta**: model params

```
# Training: Select top-K seeds based on D_train performance
seeds = [sample_seed() for _ in range(N)]
## assign sigma to each seed
sigmas_per_seed = [sigmas[i // (N // len(sigmas))]
                   for i in range(N)]

## evaluate all perturbed models
scores = [evaluate(theta + sigmas_per_seed[i] * eps(seed[i]), D_train)
          for i in range(N)]
top_indices = topk(scores, K).indices

# Inference: Ensemble predictions on test input x
answers = [generate(theta + sigmas_per_seed[i] * eps(seed[i]), x)
           for i in top_indices]
prediction = majority_vote(answers)
```

---

model is pretrained on a variety of signal shapes. Too little pretraining results in no nearby solutions and pretraining on just one signal type results in nearby weights showing very little functional diversity. See Appendix J for more examples, including generalization to held out test signals, the effect of ensembling, and settings of weight initialization and pretraining type.

## 4. A Practical Algorithm: Random Guessing & Ensembling (RandOpt)

The fact that valid solutions are easy to find but possess non-overlapping strengths, suggests that rather than searching for a single global minimum, we might instead sample broadly and aggregate the predictions. We therefore explore an algorithm, which we call RandOpt, that *randomly guesses* a set of $N$ weight perturbations, and then *ensembles* the top $K$.

Let $f_{\boldsymbol{\theta}} : \mathcal{X} \to \mathcal{Y}$ denote a base model parameterized by weights $\boldsymbol{\theta} \in \mathbb{R}^d$. We introduce perturbations via a noise vector $\boldsymbol{\epsilon}$ sampled from a standard Gaussian distribution, $\boldsymbol{\epsilon} \sim \mathcal{N}(\mathbf{0}, \mathbf{I}_d)$. The magnitude of each perturbation is controlled by a noise scale $\sigma \in \mathbb{R}^+$. To explore neighborhoods at different scales, we use a *set of scaling factors* $\Sigma = \{\sigma_1, \ldots, \sigma_M\}$. A perturbed model instance $\boldsymbol{\theta}'$ is determined by a random seed $s$ and a selected scale $\sigma \in \Sigma$:

$$\boldsymbol{\theta}' = \boldsymbol{\theta} + \sigma \cdot \boldsymbol{\epsilon}(s) \qquad (3)$$

where $\boldsymbol{\epsilon}(s)$ denotes the noise vector generated by seed $s$.

RandOpt operates in two phases: *Random Guessing* (Training) and *Ensembling* (Inference), which are given in pseudocode in Algorithm 1 and described in math below:

**Training: Random Guessing and Checking.** We sample a population of $N$ random seeds $\{s_1, \ldots, s_N\}$, and a corresponding noise scale per seed, $\{\sigma_1, \ldots, \sigma_N\}$, with all $\sigma_i$ sampled uniformly from $\Sigma$. These give a collection of $N$ parameter vectors $\theta_i = \theta + \sigma_i \boldsymbol{\epsilon}(s_i)$. Each corresponding model $f_{\boldsymbol{\theta}_i}$ is evaluated on a small training or validation set $\mathcal{D}_{\text{train}}$ to obtain a performance score $v_i$. We then select the top-$K$ performing models based on these scores:

$$\mathcal{I}_{\text{top}} = \arg \operatorname*{topK}_{i \in [N]}(v_i) \qquad (4)$$

**Inference: Ensembling of Predictions.** For a test input $x$, we generate predictions using only the selected set $\mathcal{I}_{\text{top}}$. The final output $\hat{y}$ is obtained by aggregating the individual predictions via majority voting:

$$\hat{y} = \operatorname{mode}\left( \left\{ \arg\max_y f_{\boldsymbol{\theta}_i}(y|x) \mid i \in \mathcal{I}_{\text{top}} \right\} \right) \qquad (5)$$

RandOpt differs from standard practice in several ways. First, it does not involve gradient steps, and in fact does not involve sequential updates at all – it is entirely parallel. Second, it finds a *set* of solutions, which can be ensembled, rather than a *single* setting of the weights. This latter property has also been explored in the literature on evolutionary methods and quality-diversity algorithms, which maintain a population of promising solutions rather than collapsing on a single parameter vector (e.g., Mouret & Clune (2015); Jaderberg et al. (2017); Huang et al. (2017)).

## 5. How Does RandOpt Compare to Standard Methods for Post-Training?

We test RandOpt on post-training of LLMs and VLMs, and find it to be effective across a range of settings, often

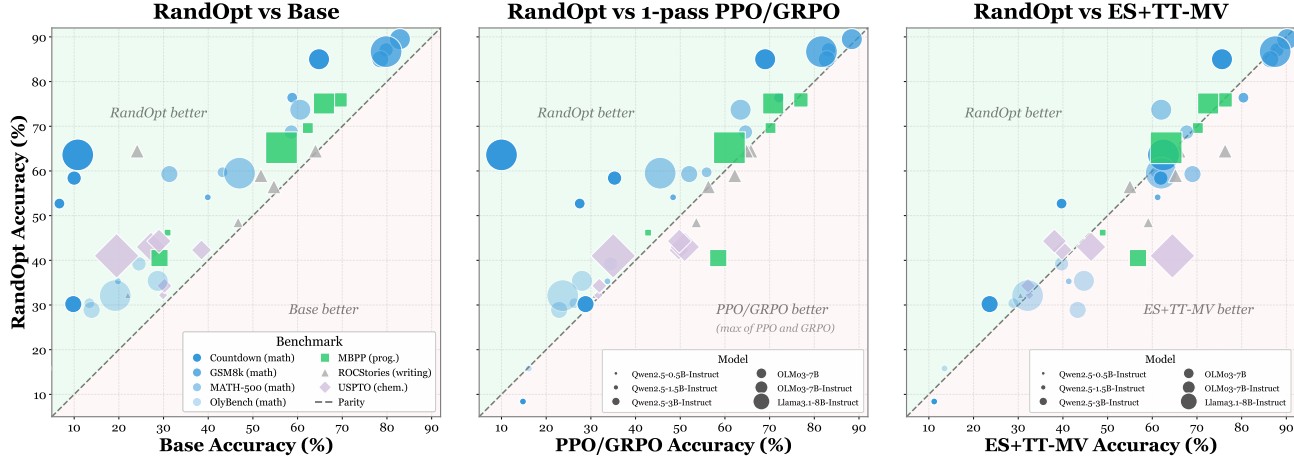

*Figure 6.* RandOpt vs. baselines on post-training LLMs. Marker size represents model scale (0.5-8B), shape indicates task family, and transparency distinguishes benchmarks within each task family. RandOpt matches or exceeds baselines in most settings. RandOpt is run with K=50 and ES+TT-MV also uses 50 test-time samples ensembled via majority vote. 1-pass PPO/GRPO use a single test-time sample; this setting disadvantages the baseline but reflects current standard usage of these models, in which there is no test-time ensembling. Full experimental results can be found in Appendix Table 4.

outperforming standard baselines. Although these results are strong, the reader should keep in mind that on these benchmarks performance can be sensitive to minor stylistic and formatting changes. Appendix F analyzes where the performance gains are coming from and argues that they are partially from reasoning improvements and partially from formatting improvements; notably, this is also true for certain baselines.

### 5.1. RandOpt on Large Language Models

We evaluate several models (Qwen, Llama, OLMo3; 0.5B–8B) covering both base and instruct variants, across four domains: math (Countdown, GSM8K, MATH-500, Oly-Bench), code (MBPP), writing (ROCStories), and chemistry (USPTO). We compare RandOpt against Test-Time Majority Vote (TT-MV), PPO, GRPO, and ES. Full details on datasets, models, and baselines are in Appendices C–E.

Our main finding is that RandOpt (with $K = 50$) mostly matches or outperforms established methods across a range of model scales (0.5B to 8B) and task categories. Benchmark performance compared to baselines, all run with equal training flops, is shown in Figure 6. More comparisons are given in Appendix H.

Notably, this performance is achieved while RandOpt involves no sequential optimization steps, whereas the baselines are run for hundreds of steps (see Appendix Table I.3 for hyperparameter settings). This gives RandOpt a potentially large wall-clock advantage, provided it is run on a large enough cluster of parallel compute. To demonstrate this, we deployed RandOpt on a 200 GH200 cluster and trained `Olmo-3-7B-Instruct` on Countdown. Using

$N = 2000, K = 50$, this takes 3.2 minutes and achieves 70% accuracy.

These experiments also reveal that the ensembling phase is crucial to RandOpt's performance. RandOpt with K=1 is substantially less effective than RandOpt with K=50, as shown in Figure 1 (c) as well as Figure 10.

### 5.2. RandOpt on Vision-Language Models

*Table 1.* RandOpt improves Accuracy (%) on GQA.

| Model | Method | GQA |
|---|---|---|
| Qwen2.5-VL-3B-Inst | Base | 56.6 |
|  | RandOpt | 69.0 |

We conduct experiments on Qwen2.5-VL-3B-Instruct, a 3B-parameter vision-language model (VLM). We evaluate on the Visual Reasoning in the Real World (GQA) dataset (Hudson & Manning, 2019), which contains questions requiring understanding of objects, attributes, and relations in images and is commonly used to benchmark visual reasoning ability. We perturb the language model while keeping the visual encoder frozen, and run RandOpt with $N = 5000$ and $K = 50$. This improves accuracy on GQA by 12.4% (Table 1).

### 5.3. Can Sandbagging Explain These Results?

Tice et al. (2025) argued that some models might be "sandbagged," where they are explicitly or implicitly trained to have low performance on certain tasks. Random weights perturbations might recover performance by breaking this effect. Can this explain our results? We think not. Most

notably, RandOpt substantially improves the OLMo3-7B Base model (see Appendix Table 4). Since OLMo's training data and recipes are open-source, we can verify that this model is free of intentional sandbagging. We provide further arguments against sandbagging as an explanation in Appendix K. However, even if sandbagging is not the right explanation, this does not mean that there could not be similarly superficial fixes that underlie the performance gains; in Appendix F we look into this in more detail and find that indeed some, but not all, of the gains are due to simply fixing answer format.

### 5.4. Can Ensembling Also Benefit the Baselines?

Yes, ensembling (e.g., 50-pass TT-MV) consistently improves baseline methods across various model scales and benchmarks. For example, in Figure 1(c), it boosts the accuracy of PPO, GRPO, and ES to approximately 79% by step 500. Table 4 also suggests this. For example, ES + TT-MV increases the GSM8k accuracy of Qwen2.5-0.5B from 42.6% to 61.2%.

In fact, ensembling benefits these models regardless of the specific selection method used during training (e.g., random guessing, GRPO, or ES). Interestingly, as training progresses, the ensembled performance gap among these different baselines gradually shrinks (Figure 1(c)).

## 6. Distillation

A catch with RandOpt, compared to standard post-training, is that good performance requires $K$ forward passes at test time. To address this, we explore distilling the top-K models into a single model.

We perform distillation on the Qwen2.5-{1.5B,3B}-Instruct models. We only use hard samples: for each input, we generate eight candidate answers and keep those for which more than half of the candidates are incorrect.

*Table 2.* Distilling the top-K RandOpt ensemble into a single model achieves performance comparable to the ensemble on GSM8K.

| Model | Method | GSM8K |
|---|---|---|
| Qwen2.5-1.5B-Inst. | Base | 58.8 |
| | Distill | 74.9 |
| | RandOpt | 76.4 |
| Qwen2.5-3B-Inst. | Base | 79.8 |
| | Distill | 84.3 |
| | RandOpt | 87.1 |

We use the top-50 models to generate 25,000 responses on 500 training examples. Each training sample is a pair $(x, [r; y])$, where $x$ is the input question, $r$ is the reasoning trace, and $y$ is the final answer. We then select hard examples and perform supervised fine-tuning (SFT) on the base model

for 2 epochs, obtaining a distilled model. Specifically, let $s = (s_1, s_2, \ldots, s_T)$ denote the full token sequence $[x; r; y]$, and let $T_x$ be the length of $x$. The SFT objective minimizes the negative log-likelihood of the reasoning trace and final answer:

$$\mathcal{L}_{\text{Distill}}(\theta) = - \sum_{t=T_x+1}^{T} \log p_\theta(s_t \mid x, s_{<t}), \quad (6)$$

where $\theta$ denotes the model parameters. The input question $x$ is the context (with its loss masked), and the model learns to autoregressively generate the reasoning trace and final answer $[r; y]$.

The computational cost of distillation is small compared to training. Since training uses a population size of 5,000, and distillation only uses the top-50 models and runs for 10 SGD iterations, the cost of distillation is about 2% of the training cost.

## 7. Related Work

### 7.1. Structure of the Neural Net Loss Landscape

**Flat Minima**    A prominent finding in the loss landscape literature is that training tends toward flat minima (Keskar et al., 2017). Our findings reveal that flat minima can be hiding important structure below the surface: *per-task*, the local accuracy landscape is not nearly so flat and the pretrained weights can even lie in a trough of accuracy. Pretraining aggregates over many tasks, hence a flat pretraining landscape is compatible with spiky per-task losses.

**Multi-Task Loss Landscapes**    A large body of work aims to characterize the loss landscape of neural nets trained for a single objective (e.g,. Li et al. (2018); Choromanska et al. (2015)). Thickets, in contrast, are a property of the multi-task landscape. Prior work that takes a similar perspective includes Pareto front learning, where paths in weight space are identified that tradeoff between different task objectives (Ma et al., 2020), and multi-task linear mode connectivity, where linear paths of low loss are observed between the minimizers of different tasks (Mirzadeh et al., 2020).

**On Lottery Tickets and Neural Thickets**    The Lottery Ticket Hypothesis suggests that, when training from scratch, finding a good initialization is akin to winning the lottery: random initialization will only rarely sample weights that train well (Frankle & Carbin, 2019). Our findings are compatible with this view, but suggest a qualitatively different regime after pretraining. At transfer time, the neighborhood around the initialization (i.e. around the pretrained weights) becomes abundant with good solutions.

## 7.2. Post-Training as Selection

**Reweighting the Pretrained Policy** Many works have observed that post-training methods can be interpreted as reweighting behaviors already present in the pretrained distribution. For example, KL-regularized methods such as PPO (Schulman et al., 2017) keep the policy close to the pretrained model, and can be interpreted as reweighting the pretrained distribution (Rafailov et al., 2023).

**Self-improvement by Trace Selection** In the self-improvement literature, a common recipe is to use test-time search to select good reasoning traces, and then train them back into the model weights (e.g., Zelikman et al. (2022); Xiong et al. (2025)). These approaches aim to convert high pass@k performance into high pass@1 performance. Our results are consistent with the view that post-training selects or sharpens skills that are already latent in the pretrained model. While prior work characterizes how probability mass can be reweighted in output-space, we instead characterize the geometry of nearby weight-space optima.

## 7.3. Low-dimensional structure in LLM fine-tuning

**Intrinsic dimension and parameter-efficient fine-tuning.** Prior work (Aghajanyan et al., 2020) shows that fine-tuning often succeeds within a surprisingly small random subspace, suggesting that downstream adaptation is effectively low-dimensional despite the enormous parameter space of LLMs. Consistent with this view, parameter-efficient fine-tuning methods such as LoRA (Hu et al., 2022) restrict updates to low-rank components while freezing the base model, yet still achieve competitive performance across many tasks. More recently, (Morris et al., 2026) showed that math reasoning tasks can be learned by updating only 13 parameters.

**Low-dimensional curvature in LLM fine-tuning.** (Liang et al., 2026) shows that LLM fine-tuning landscapes exhibit *low-dimensional curvature*, where a small number of directions dominate reward improvements. Random projections have a higher chance of intersecting with a large, degenerate set of reward-improving directions as a consequence of the low-dimensionality. This view suggests interpreting the thickets phenomenon as the intersection of (a) a broad loss basin induced by pretraining and overparameterization, and (b) a set of task-relevant directions that are effectively low-dimensional (or low-rank) but embedded within the full parameter space.

# 8. Implications

## 8.1. Rethinking Pretraining

**Pretrained Models as Distributions** We typically refer to the "pretrained model" as a singular thing; it's *the* base model, or it's *a* foundation model, on top of which further improvements can be made. Our results suggest that you can instead think about your pretrained weights as specifying a distribution over models. This distribution resists characterization just in terms of its mean: rather it contains diverse specialists whose behavior is qualitatively different from the singular pretrained weights.

**Understanding Pretraining Requires Characterizing the Multi-task Loss Landscape** While the pretraining objective necessarily exhibits a local minimum at converged weights, our findings suggest that this scalar landscape obscures important structure. In particular, what governs downstream adaptation is not a single loss surface, but the collection of task-specific loss landscapes corresponding to different downstream objectives. The pretrained weights might not be a minimizer of any individual element of this collection, but lie in a region surrounded by task-specific minima. This structure is invisible in the aggregate pretraining objective, and only emerges when that objective is decomposed into its constituent task-level losses.

## 8.2. Rethinking Post-Training

**Pretraining Is All You Need?** We have shown that once a model has been sufficiently pretrained, further adaptation can be remarkably easy. This finding mirrors prior work, including: Tian et al. (2020) found that linear probes over pretrained representations can outperform sophisticated meta-learners; Finn & Levine (2018) proved that, given a good enough representation, gradient descent can approximate any learning algorithm; Qiu et al. (2025) showed that relatively simple algorithms such as ES can rival state-of-the-art RL methods for post-training LLMs. These are a few of the papers that, together with our work, suggest that it doesn't take much to obtain good downstream solutions given the right pretrained representation.

**Decentralized, Parallel Adaptation** RandOpt workers operate fully in parallel, and do not communicate with each other during training. Only at inference time do the workers interact, and only through ensembling their predictions. This may be attractive in a setting where compute nodes are cheap but communication is at a premium. Prior work has argued that ES requires less communication bandwidth than certain RL methods (Salimans et al., 2017), and RandOpt is cheaper still: $T$ steps of ES requires communicating scores $T$ times, while RandOpt requires communicating scores just once. Further, RandOpt could be preferable where wall-clock time is what matters: RandOpt's wall-clock time is $\mathcal{O}(1)$ in optimization steps whereas sequential methods like ES are $\mathcal{O}(T)$. Due to its decentralized nature, RandOpt may also be especially suitable for federated settings where data security or privacy are paramount.

## Impact Statement

This work has both scientific implications on our understanding of foundation models, and may have practical impact toward better post-training methods. Implications toward both theory and practice are discussed in the main paper.

## Acknowledgements

This work was supported under project ID 43 as part of the Swiss AI Initiative, through a grant from the ETH Domain and computational resources provided by the Swiss National Supercomputing Centre (CSCS) under the Alps infrastructure. This work was also supported by a Packard Fellowship to P.I., and a Frederick (1953) and Barbara Cronin Fellowship to Y.G., and by ONR MURI grant N00014-22-1-2740. We thank Minyoung Huh and Jeremy Bernstein for inspiring discussions on earlier iterations of this project.

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

# A. Additional Related Work

## A.1. Randomized Search and Evolutionary Methods

**Random Search Can Be Effective for Training and Inference with Neural Nets**    Many prior papers have shown that *sequential* random search methods are competitive with RL for control problems (e.g., Salimans et al. (2017); Mania et al. (2018)) and even for post-training LLMs (Qiu et al., 2025). However, we are not aware of prior work that demonstrated the same for the parallel search case, except in very simple scenarios, such as those explored by Schmidhuber et al. (2001) and by Oller et al. (2020). Parallel guess-and-check methods, such as Best-of-N, are also commonly used at test-time to improve model performance, and these methods perform well compared to more sophisticated inference methods (Wu et al., 2025). We note that the training phase of RandOpt is essentially Best-of-N in weight-space, rather than in output space. Given a verifier or reward signal, RandOpt could be applied at test-time as well.

**Spurious Rewards Can Sometimes Be Effective**    For some tasks, we find that density is so high that *most* Gaussian perturbations increase task accuracy (Figure 12). This phenomenon may provide a partial explanation of the recent finding that post-training on random or spurious rewards can sometimes be effective (Shao et al., 2025). Such rewards provide gradients in the wrong direction, but this wrong direction might still, by chance, be sufficiently right.

**Evolving Adaptable Initializations**    The central conceit of our paper is that pretraining draws weights toward regions surrounded by adaptive specialists. Simpson (1953) argued that this property also holds true for evolved genomes, referring to this as the "Baldwin Effect," in reference to prior work by James Baldwin (Baldwin, 1896). Hinton et al. (1987) provided an initial computational model of this effect. Later, (Gupta et al., 2021; Feng et al., 2025) simulated this effect in embodied agents. In short, evolution tends toward inits from which within-life learning can quickly adapt. These works provide a backdrop for modern methods in meta-learning, which optimize for neural net initializations from which task-specific solutions are a short step away. Prominent in this family is the MAML algorithm of Finn et al. (2017). Our results indicate that pretraining is implicitly finding a MAML-like init.

## A.2. Direct Models of Weight Space

**Bayesian Neural Nets and Parameter Noise**    Bayesian neural nets treat parameters as random variables, which can be sampled from to estimate distributions over outputs (Goan & Fookes, 2020). This approach is often used to quantify uncertainty and calibrate predictions, or to improve predictions by ensembling over samples (Gal & Ghahramani, 2016). Our new observation is that pretrained weights can be usefully treated as Gaussian random variables even when they were not trained to have this property. In other words, we view pretrained nets as *implicitly* defining a distribution of representations about their weights. This differs from prior works on Bayesian methods that explicitly represent these distributions. Of prior approaches, PEP (Mehrtash et al., 2020) is especially close to RandOpt. PEP computes an ensemble of predictions from Gaussian perturbations of model weights; unlike RandOpt, however, PEP has no selection step, aside from optimizing the variance of the Gaussian.

**Weight Space Model Editing**    While most learning methods manipulate weights indirectly, e.g., by backpropagating errors, there is also work on directly manipulating weights. Cherepkov et al. (2021) found that linear directions in the weight space of a generative adversarial network map to interpretable edits of the generated image; this is a weight-space analog to the popular notion that activation-space admits interpretable linear edits (Park et al., 2024). Dravid et al. (2024) found that similarly simple weight manipulations work for diffusion models. More broadly, low-rank weight manipulations have become especially popular for model editing (Hu et al., 2022), as we discuss more next. Collectively, these works suggest that in weight-space, meaningful adaptations require only minor changes. The thickets phenomenon could help explain why.

## A.3. Parameter-space exploration

A related line of work studies parameter-space perturbation as a mechanism for exploration in reinforcement learning. Fortunato et al. (2019); Plappert et al. (2017) inject noise into model weights and learn the noise parameters jointly with the base model to enable more efficient learning and exploration. In these methods, perturbations are used during learning to improve exploration or optimization. Our framing is different: we ask whether perturbations around pretrained weights can find nearby task experts after pretraining.

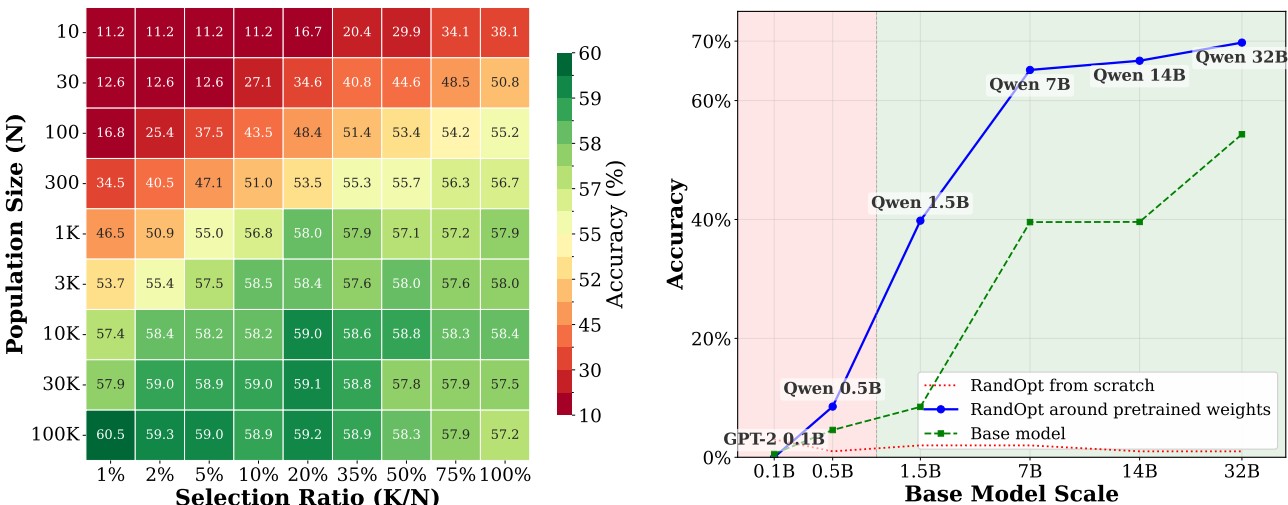

*Figure 7.* Heatmap of accuracy across population size $N$ and selection ratio $K/N$. Accuracy scales with population size. Task: Countdown, Model: Qwen2.5-3B-Inst.

*Figure 8.* Relationship between model scale and RandOpt performance. Good pretrained representations are important for RandOpt start to work. Task: Countdown, N = 3k, K = 50.

## B. Scaling Properties of RandOpt

We investigate how RandOpt performance scales with respect to the search budget (population size $N$), the ensemble size ($K$), and the base model size.

**Impact of Population Size and Selection Ratio.** Figure 7 measures the interplay between search budget ($N$) and selection ratio ($K/N$) on the Countdown task, with Qwen2.5-3B-Inst (see Appendix Figure 9 for further slices of this data). Note that the precise shape of these trends may vary from task to task and model to model, and here we only show results for Countdown. Two observations:

(1) For sufficiently low selection ratio, performance improves monotonically with population size $N$.

(2) Trade-off between $N$ and $K/N$: the optimal selection ratio decreases with increasing population size.

Note that the top-1 model's performance on the training set is, by construction, non-decreasing in $N$. This is consistent with observation #1; if a practitioner wants to avoid hyperparameter tuning, a reasonable strategy would therefore be to set $K$ small and set $N$ as large as their budget can afford.

**The Emergence of Thickets at Scale.** Figure 8 illustrates the relationship between base model scale and RandOpt's performance, showing that a strong base model is essential for RandOpt to start to work. The blue line corresponds to applying RandOpt to the base models, and the green dashed line is the performance of the base models. For very small models, such as GPT-2 0.1B, RandOpt fails to improve performance; for small models (e.g., Qwen 0.5B), RandOpt also offers small gains over the baseline. However, starting at around 1.5B parameters, RandOpt triggers a rapid increase in accuracy. After this point, the base model accuracy begins to catch up and the relative improvement of RandOpt shrinks as the models plateau. Using RandOpt on a model that has not been pretrained ("RandOpt from scratch," red dotted line) remains near zero across all scales. These results suggest that sufficient pretraining and sufficient model scale are essential for RandOpt to work; this matches our finding in Section 2 for the conditions under which thickets emerge.

## C. Models

We conduct experiments across the Qwen, Llama, and OLMo3 model families. Our selection encompasses diverse model sizes (0.5B to 8B parameters), multiple model families with different pretraining recipes, and both instruction-tuned and base models. Specifically, we evaluate:

*Qwen2.5-Instruct* (Qwen et al., 2025) at three scales (0.5B, 1.5B, and 3B). Qwen2.5-Instruct is a series of instruction-tuned

language models ranging from 0.5B to 72B parameters, demonstrating strong performance on reasoning and coding tasks.

*Llama-3.1-8B Instruct* (Grattafiori et al., 2024). This is an instruction-tuned variant of the Llama 3.1 family, optimized for dialogue and instruction-following capabilities.

*OLMo3* (Olmo et al., 2025) in both base and instruction-tuned variants at 7B. We include OLMo3 as it is fully open-source with transparent training data and procedures, mitigating concerns about potential data contamination or sandbagging that may affect evaluation integrity.

## D. Datasets

We evaluate our method on benchmarks spanning five task categories: **mathematical reasoning** (Countdown, GSM8K, OlympiadBench, MATH-500), **code generation** (MBPP), **creative writing** (CommonGen), **chemistry** (USPTO) and **commonsense** (GQA), a visual question answering benchmark commonly evaluated with vision-language models (VLMs).

**Mathematical Reasoning.** *Countdown task* (Countdown, 2024; Gandhi et al., 2024) measures symbolic and numerical reasoning ability by requiring models to construct arithmetic expressions that exactly reach a target value given a set of numbers.

*GSM8K* (Cobbe et al., 2021) is a widely used benchmark for grade-school–level mathematical reasoning, consisting of multi-step word problems that require arithmetic calculations and logical reasoning.

*OlympiadBench* (He et al., 2024) is a bilingual benchmark consisting of 8,476 Olympiad-level mathematics and physics problems drawn from international and Chinese competitions, designed to evaluate scientific reasoning capabilities including theorem application, multi-step derivations, and complex problem solving.

*MATH-500* (Mayilvahanan et al., 2025) is a challenging subset of the MATH dataset, focusing on competition-level mathematical problems that test advanced multi-step reasoning and symbolic manipulation.

**Code Generation.** *MBPP* (Austin et al., 2021) is a benchmark of approximately 1,000 crowd-sourced Python programming problems designed to be solvable by entry-level programmers. Each problem consists of a task description, a reference code solution, and three automated test cases, covering programming fundamentals and standard library functionality to evaluate function-level code generation capabilities.

**Creative Writing.** *ROCStories* (Mostafazadeh et al., 2016) is a commonsense narrative generation benchmark consisting of short everyday stories. The dataset contains around 100,000 five-sentence narratives describing real-world events. It evaluates a model's ability to generate coherent, fluent, and logically consistent story continuations grounded in commonsense reasoning.

**Chemistry.** *USPTO* (van der Lingen, 2023) is a large-scale chemical reaction dataset extracted from United States Patent and Trademark Office patent documents, containing over 1.8 million organic chemical reactions represented as reaction SMILES. The benchmark evaluates models on reaction prediction and retrosynthesis tasks, requiring understanding of chemical transformations and molecular structure relationships.

**Commonsense.** *GQA* (Hudson & Manning, 2019) is a visual question answering benchmark designed to evaluate compositional visual reasoning and grounded commonsense understanding. Questions require models to perform multi-step reasoning over object attributes and relations (e.g., spatial relationships, colors, or object interactions), testing the ability to combine visual grounding with commonsense reasoning.

## E. Baseline Methods

**Test-Time Majority Vote.** Majority voting (Wang et al., 2023) is a test-time inference strategy that samples multiple independent responses from a model and selects the most frequently occurring answer. This approach improves accuracy without updating model parameters by leveraging the diversity of sampled reasoning paths.

**Best-of-N.** Best-of-N (Ouyang et al., 2022) samples multiple responses at inference time and selects the highest-scoring one according to a predefined evaluation metric or reward function, rather than aggregating answers by frequency as in majority voting.

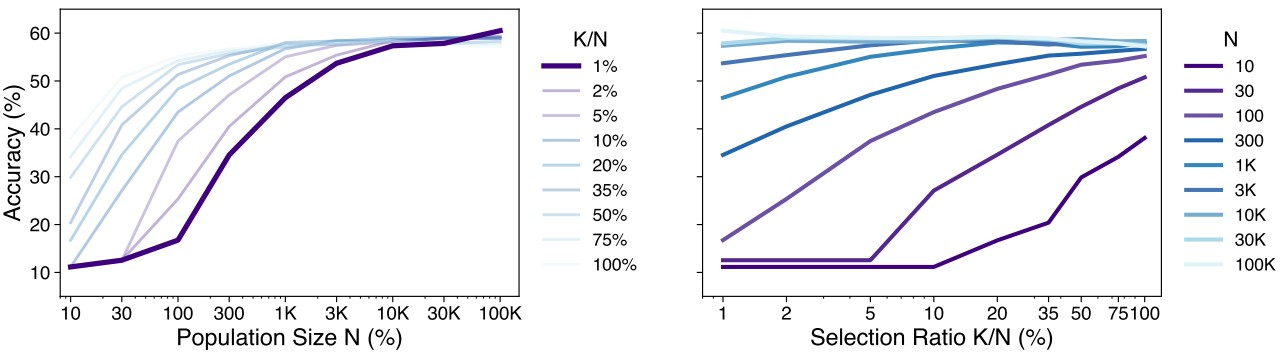

*Figure 9.* Scaling curves for population size $K$ and ensemble size $N$ on Countdown task.

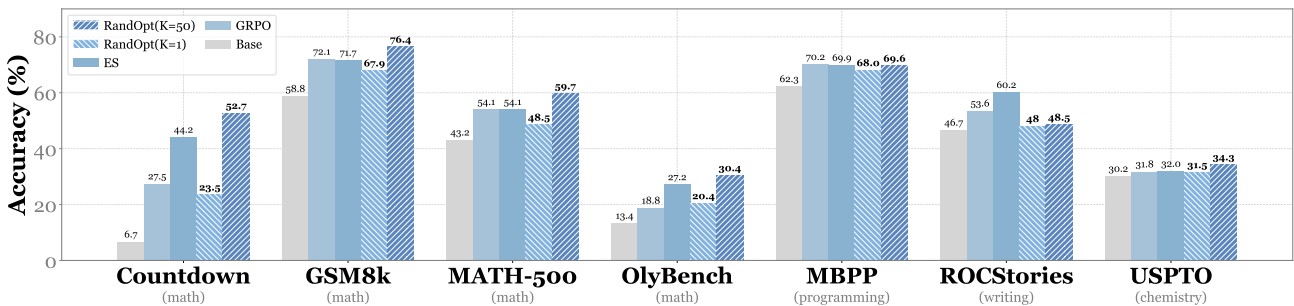

*Figure 10.* Model performance comparison on Countdown, GSM8k, MATH-500, and OlympiadBench using the Qwen2.5-1.5B-Instruct model. More results for additional models and baselines can be found in Appendix Table 4.

**PPO.** Proximal Policy Optimization (Schulman et al., 2017) uses a clipped surrogate objective to stabilize policy updates in RLHF, requiring both a policy model and a critic network.

**GRPO.** Group Relative Policy Optimization (Shao et al., 2024) removes the critic by computing advantages from group-level reward statistics, reducing memory overhead compared to PPO.

**ES.** Evolution Strategies at Scale (Qiu et al., 2025) perform gradient-free optimization by perturbing parameters with Gaussian noise and updating based on fitness-weighted perturbations.

## F. Types of Thickets

It is possible that the effects we have observed do not arise from models in the thicket employing fundamentally different forms of reasoning, but instead from differences that are comparatively shallow. For example, models may vary instead in surface-level behaviors such as answer formatting or style. Task performance can be highly sensitive to such factors: a system that expects outputs in JSON may fail entirely if a model instead emits free-form text. Are our results simply due to random perturbations improving output formatting?

We test this by measuring how much of the improvement on GSM8K is attributable to formatting fixes versus correcting the actual numerical answer. On 1319 test data samples, we decompose performance, relative to the base model, into 1) retained correctness, 2) base wrong, adapted model correct (indicating a "reasoning thicket", where perturbations can help the model solve reasoning problems it could not solve before), 3) format fixed, then counted as correct (indicating a "format thicket", where the base model solved the problem but output the answer in a format that was marked as incorrect by a strict answer checker, e.g., the answer was not placed after the proper tag "####"), and 4) base correct, adapted model incorrect (a regression in model ability).

Results are shown in Figure 11. RandOpt ($K = 50$), reaches 86.7% overall accuracy with 0.7% regression, while still showing substantial contributions from both format (19.0%) and reasoning (12.3%) thickets.

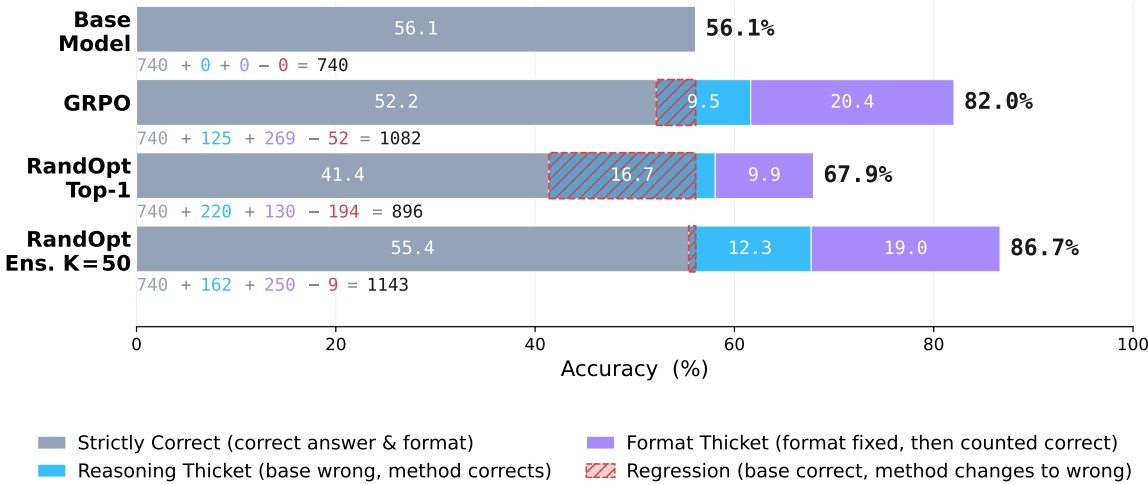

*Figure 11.* Accuracy decomposition on GSM8k using Qwen2.5-3B-Instruct (N=3000, K=50). Grey denotes strictly correct answers (both format and answer correct), light blue denotes the questions originally answered incorrectly that are corrected after training, and purple denotes the gains are just from fixing format. For both GRPO and RandOpt, a large portion of the gains come from format correction, while another portion is from solving the problem correctly.

This experiment demonstrates that thickets can come in a variety of types: we might have thickets of different answer formats, thickets of reasoning approaches, thickets of personalities, thickets of domain knowledge, and more. All of these in combination could contribute to task experts being dense and diverse, since to be an expert at a task, as defined in this paper, simply requires *doing well on the benchmark for that task* and benchmarks measure a combination of format, skill, personality, knowledge, and more. Beyond language models, in Appendix N, we show a case of "color thickets" in an image generative model.

## G. Limitations

**Pretraining Might Be All You Need, But You Do Need Pretraining** RandOpt is not suitable for training large neural nets from scratch, and achieves negligible performance in this setting (e.g., see the dotted red line in Figure 8). It also struggles on small pretrained models, where the density of solutions is low. The success of RandOpt around well-pretrained inits does not mean we should discard other learning algorithms. Rather RandOpt works once you have a good enough representation, but to find that representation in the first place may still require structured search.

**Capacity to Learn Dramatically New Skills?** Our results leave open the question of exactly how far beyond the base model's abilities random guessing and ensembling can take us. The scaling relationships we observe appear to saturate at large model size (Figure 9) and large $N$ (Figure 7); the saturation is visible even as a function of log resources. This may indicate that further improvement on these tasks requires exiting the local thickets and hunting farther and wider, where the search might return to needle-in-haystack dynamics and more structured methods may be necessary.

**Inference-Time Cost** At inference time RandOpt uses $K$ forward passes, and good performance typically requires $K > 1$. This cost can be reduced by distilling the ensemble into a single model, as we have shown in Section 6. However, distilling introduces several tradeoffs: 1) the algorithm is no longer fully parallel, 2) distillation requires additional training flops (although we found this cost to be minor in our experiments), and 3) our specific distillation approach is tailored to LLM reasoning toward a categorical final prediction; this approach might not be applicable in other settings.

**Majority-Vote Ensembling Does Not Support Structured Prediction** We have focused primarily on problems where the answer is a single discrete class (or an integer), in which case majority-vote ensembling is straightforward to apply. In the 1D signal experiments in Appendix J, we also show a case where mean ensembling can work. It is less clear how to ensemble, or distill, models that perform more structured kinds of prediction, such as writing a story, generating an image, designing a molecule, etc. To handle these cases, the ensembling approach in Algorithm 1 would need to be modified. In

Appendix N, we show one simple ensembling approach on image generation with a diffusion model: mean ensembling at each step of denoising. We do not claim that this is the best choice but rather include it as a proof of concept that our framework could be extended to many possible ensembling methods beyond just majority voting.

**Exactly When and Why Does Pretraining Enter the Thicket Regime?**  Our results characterize properties of the pretrained landscape, but do not fully explain the mechanisms by which these properties arise. The experiments in Section 3 show a setting in which pretraining on a distribution of many different tasks is critical to thickets forming. Is this also the critical factor in developing thickets in LLMs and other large models? What exactly is it about the pretraining objective, or learning dynamics, that creates thickets? Our results invite further investigation.

## H. Additional Experimental Analysis

**Full results on LLM**  For post-training LLMs, Table 4 reports our full results across different base models, adaptation methods, and tasks.

**Effect of ensembling**  Figure 10 compares RandOpt (K=1) to RandOpt (K=50), alongside several baselines. These results show that ensembling over many perturbations is critical to getting competitive performance on most tasks, but also that even without ensembling (just taking the top perturbation; K=1), we observe a substantial performance boost over the base model.

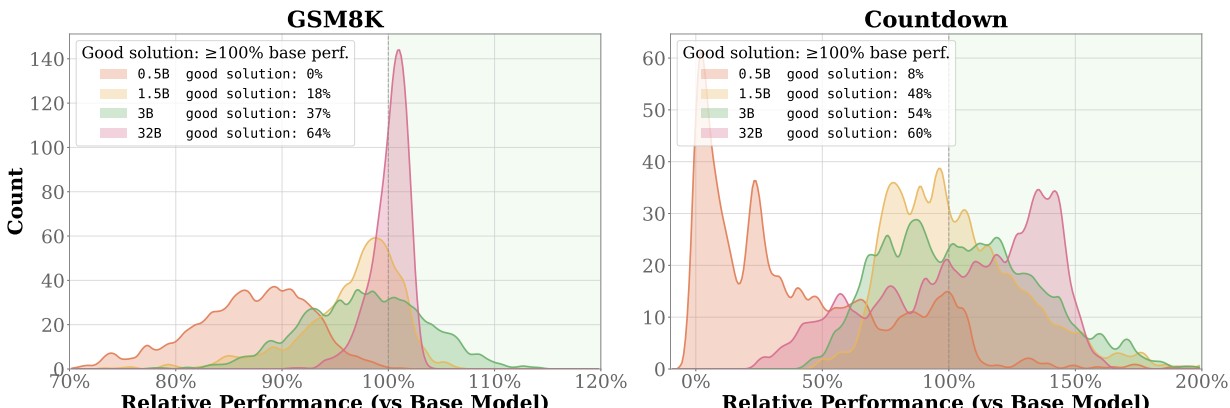

*Figure 12.* Performance distributions of 500 randomly perturbed models relative to base model accuracy on the math reasoning task GSM8k (left) and Countdown (right). The fraction of perturbations matching or exceeding base performance (shaded region) increases with model size: $0\% \rightarrow 64\%$ on GSM8K and $8\% \rightarrow 60\%$ on Countdown as model size grows from 0.5B to 32B parameters. Note: x-axis shows relative performance; longer tails in smaller models do not equate to higher absolute accuracy.

**Solution density histograms**  In Figure 12, we show the full distribution of performance improvement over random weight perturbations for GSM8K and Countdown, on the Qwen2.5-Instruct series of models.

**Log-linear correlation between population size ($N$) and performance.**  Appendix Figure 9 illustrates the scaling properties of our method on the Countdown task. As shown in the left panel, we observe a log-linear correlation between population size and performance across different selection ratios. RandOpt benefits significantly from scaling $N$ in the population size.

**Selection ratio saturation.**  The right panel shows that while increasing the selection ratio significantly improves performance when $N$ is small, this benefit saturates as $N$ scales up. For sufficiently large $N$, the minimal value of $K$ (selecting only the top 1%) yields high performance. This demonstrates that a large population size $N$ allows for a very small topK selection, which can reduce inference time costs.

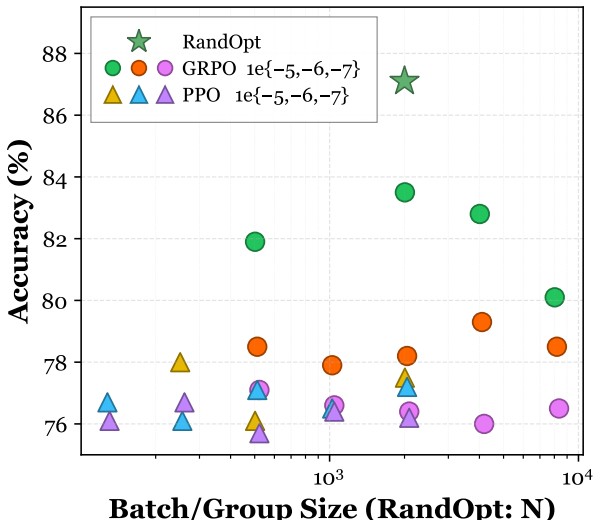

*Figure 13.* Batch/group size vs. accuracy under one-step training. Scaling baseline parallelism does not get RandOpt's performance. Task: GSM8K, Model: Qwen2.5-3B-Instruct, N=5000, K=50)

**Scaling Parallelism of the Baselines Does Not Match RandOpt (1-Step Training).** Figure 13 plots batch/group size versus accuracy on GSM8K using Qwen2.5-3B-Instruct, with all methods run for one training step. The key takeaway is that using larger batch/group size does not really help. We grid-search GRPO and PPO on 8 GPUs over learning rates $1e-5$, $1e-6$, and $1e-7$; batch size (PPO) ranges from 128 to 2048, and group size (GRPO) ranges from 512 to 8192. The x-axis is batch/group size (RandOpt uses population size $N$), color encodes learning rate, and marker shape denotes algorithm.

**PPO** does not benefit consistently from larger batches: best accuracy is 78.0% at batch size 256 (lr=$1e-5$), while larger-batch runs such as batch size 2048 reach at most 77.5%.

**GRPO** reaches its peak at 83.5% (group size=$512 \times 4$, lr=$1e-5$), but increasing group size does not improve performance (e.g., 80.1% at $2048 \times 4$ with lr=$1e-5$).

**RandOpt** ($N = 3000$) reaches 87.1%, exceeding all GRPO and PPO settings. This indicates that scaling baseline parallelism does not close the gap with RandOpt 's performance.

## I. Implementation Details

All experiments are conducted on NVIDIA GH200 GPUs. To ensure a fair comparison, we normalize the computational budgets of all methods by the total training FLOPs. For the experiments in Table 4, We have two variants that differ in their selection strategy: RandOpt (random) performs selection by random sampling, while RandOpt (ES) updates the population using an evolution strategies (ES) algorithm and selects the top-50 models from the final ES population. We use a population size of 5,000 and ensemble the top-50 models for RandOpt (random), and a population size of 100 with 50 iterations and ensemble the top-50 models for RandOpt (ES). For the figures in Section 2, we use $\sigma = 0.001$. For all datasets, we use the first 200 samples from the training set as the RandOpt training set, and all samples from the test set for evaluation. For MATH-500, we manually split the dataset: the first 200 samples are used for training and the remaining samples for testing. For OlympiadBench, we use the text-only English competition math subset (`OE_TO_maths_en_COMP.parquet`), since most other subsets require visual inputs.

### I.1. FLOPs Calculation

For a model with $P$ parameters and a sequence length of $L$, we follow the standard estimation where a single forward pass requires approximately $2PL$ FLOPs, and a backward pass requires $4PL$ FLOPs (Narayanan et al., 2021). The total training compute is determined by the number of samples processed and the computational overhead per sample.

### I.2. Method-Specific Compute

- **GRPO**: Each step processes a batch of $B$ questions with $G$ responses per question. Each response involves a policy forward, a reference model forward, and a policy backward pass.

$$\text{FLOPs}_{\text{GRPO}} = T_{\text{GRPO}} \cdot B \cdot G \cdot \underbrace{(2 + 2 + 4)}_{\text{fwd + ref + bwd}} \cdot PL = \mathbf{8} \cdot \mathbf{T_{\text{GRPO}}} \cdot \mathbf{B} \cdot \mathbf{G} \cdot \mathbf{PL} \tag{7}$$

- **PPO**: Adds critic forward and backward passes to the GRPO baseline.

$$\text{FLOPs}_{\text{PPO}} = T_{\text{PPO}} \cdot B \cdot G \cdot \underbrace{(2 + 2 + 2 + 4 + 4)}_{\text{fwd + ref + crit\_fwd + bwd + crit\_bwd}} \cdot PL = \mathbf{14} \cdot \mathbf{T_{\text{PPO}}} \cdot \mathbf{B} \cdot \mathbf{G} \cdot \mathbf{PL} \tag{8}$$

- **ES & RandOpt**: These gradient-free methods only require forward passes for evaluation. For a population size $N$ and

evaluation dataset size $D$:

$$\text{FLOPs}_{\text{ES/RandOpt}} = T_{\text{ES}} \cdot N \cdot D \cdot \underbrace{2}_{\text{fwd}} \cdot PL = \mathbf{2} \cdot \mathbf{T_{\text{ES}}} \cdot \mathbf{N} \cdot \mathbf{D} \cdot \mathbf{PL} \tag{9}$$

### I.3. Hyperparameters

Appendix Table 3 shows the hyperparameters across methods. To ensure a fair comparison, we align the hyperparameters such that all methods consume equivalent total training FLOPs. We balance the batch size and iteration counts to account for algorithmic overheads. For instance, GRPO uses a larger batch size ($B = 1024$) compared to PPO ($B = 128$) due to the latter's additional memory cost for the critic network. Similarly, we match the total number of sample evaluations between the iterative ES (30 population $\times$ 167 steps) and the single-step RandOpt (5000 population $\times$ 1 step). All experiments utilize bfloat16 precision with a maximum sequence length of 1024.

*Table 3.* Hyperparameters. '–' indicates that the hyperparameter is not required for the corresponding algorithm.

| Category | Parameter | GRPO | PPO | ES | RandOpt |
|---|---|---|---|---|---|
| Budget & Scale | Batch Size | 1024 | 128 | – | – |
| | Group/Population Size | 8 | 1 | 30 | 5000 |
| | Iterations | 200 | 600 | 167 | 1 |
| Optimization Complexity | Actor Learning Rate | 1e-6 | 1e-6 | 5e-4 ($\alpha$) | – |
| | Critic Learning Rate | – | 1e-5 | – | – |
| | Optimizer | AdamW | AdamW | - | – |
| | KL Coefficient ($\beta$) | 0.001 | 0.001 | – | – |
| | Backpropagation | Yes | Yes | – | – |
| Model Requirements | Reference Model | Required | Required | – | – |
| | Critic Model | – | Required | – | – |
| | Value Head | – | Required | – | – |
| Other Hyperparams | Perturbation Scale ($\sigma$) | – | – | 0.001 | {1, 2, 3}e-3 |
| | Precision | bf16 | bf16 | bf16 | bf16 |
| | Max Seq. Length | 1024 | 1024 | 1024 | 1024 |

## J. Additional Results on 1D Signals

We show additional results of the experiments on 1D signals in Tables 5 and 6. Table 5 tests the approximation ability of RandOpt: here we select perturbations based on their ability to fit a single test function (plotted in blue). Table 6 instead tests generalization: we select the top-K perturbations that perform best on a post-training dataset, then plot predictions on a newly sampled test function of the same function type as the post-training type (noted in the second column). In this table, we also include mean performance over 256 randomly sampled test functions of the same type (rightmost column).

All three of the regimes we have discussed in the main paper can be observed here:

1) *Needle in haystack* regime: Without pretraining (Table 5, top two rows), the Gaussian perturbations do not sample effective models. We show this for two different weight initializations: Xavier initialization (Glorot & Bengio, 2010) and Kaiming initialization (He et al., 2015). In both cases, we need to ramp up the $\sigma$ value of the perturbations to see any visible effect. Nonetheless, even with large enough $\sigma$ to see interesting variation in the predictions, the variation does not contain good continuations of the test functions.

2) *Thicket* regime: With pretraining on mixed signal types (Table 5, row 3, Table 6 rows 1-3), the base model fails to reliably predict the correct type given a limited context; after RandOpt post-training on this type, the results improve.

3) *Plateau* regime: If you pretrain on just a single function type and test on this type (Table 5 bottom row, left column; Table 6 rows 4-5), then the base model already is at ceiling performance and further adaptation is unnecessary.

## K. Additional discussion on Sandbagging

In this section, we provide a more detailed discussion on why we think the performance improvements of RandOpt cannot be attributed to the alleviation of "sandbagging." Sandbagging refers to a phenomenon where a model underperforms relative to its true underlying capabilities, perhaps as a side-effect of safety alignment or instruction tuning that penalizes certain types of outputs. A potential concern is that if the baseline model's performance is artificially suppressed by such alignment, any perturbation might simply bypass these constraints to reveal pre-existing latent performance.

**Evidence from Transparent Base Models.** The strongest evidence against the sandbagging hypothesis comes from our results on the OLMo3-7B Base model. Unlike many proprietary models whose training recipes are unknown, OLMo's training data, code, and training pipeline are open-source. This transparency allows us to verify that the base model is not subject to explicit sandbagging. As shown in Table 4, RandOpt is still quite effective on this model.

**Inconsistency with Strategic Behavior in Small Models.** Sandbagging is generally considered a property of larger models. However, RandOpt yields consistent gains across all scales. For instance, on the Qwen2.5-0.5B-Inst model, RandOpt improves GSM8k performance from 39.9% (Base) to 61.2%.

**Comparison with Test-Time Majority Voting (TT-MV).** If the models were simply sandbagging by occasionally providing wrong answers despite "knowing" the correct ones, test-time techniques like Majority Voting (TT-MV) might effectively recover that latent performance. Our experimental results in Table 4 show that RandOpt consistently outperforms TT-MV across most benchmarks.

## L. Derivation of Spectral Discordance Bounds

Here we provide the formal proof for the theoretical bounds of the Spectral Discordance metric $\mathcal{D}$ defined in the main text.

**Proposition L.1.** *For any valid correlation matrix $\mathbf{C} \in \mathbb{R}^{M \times M}$, the Spectral Discordance $\mathcal{D} = 1 - \frac{1}{M(M-1)} \sum_{j \neq k} \mathbf{C}_{jk}$ is bounded by:*

$$0 \leq \mathcal{D} \leq \frac{M}{M-1} \tag{10}$$

*Proof.* Let $\bar{\rho}$ denote the average off-diagonal correlation:

$$\bar{\rho} = \frac{1}{M(M-1)} \sum_{j \neq k} \mathbf{C}_{jk} \tag{11}$$

By definition, $\mathcal{D} = 1 - \bar{\rho}$. We analyze the bounds of $\bar{\rho}$.

**1. Lower Bound of $\mathcal{D}$ (Upper Bound of $\bar{\rho}$):** The maximum value of any correlation coefficient is 1. If all tasks are perfectly correlated ($\mathbf{C}_{jk} = 1, \forall j, k$), then $\bar{\rho} = 1$, yielding the minimum discordance:

$$\mathcal{D}_{\min} = 1 - 1 = 0 \tag{12}$$

This corresponds to the *Generalist* regime where all tasks ranks are identical.

**2. Upper Bound of $\mathcal{D}$ (Lower Bound of $\bar{\rho}$):** The correlation matrix $\mathbf{C}$ must be positive semi-definite (PSD). Consider the quadratic form with the all-ones vector $\mathbf{1} \in \mathbb{R}^M$:

$$\mathbf{1}^\top \mathbf{C} \mathbf{1} \geq 0 \tag{13}$$

Expanding this quadratic form:

$$\sum_{i=1}^{M} \sum_{j=1}^{M} \mathbf{C}_{ij} = \sum_{i=1}^{M} \mathbf{C}_{ii} + \sum_{j \neq k} \mathbf{C}_{jk} \geq 0 \tag{14}$$

Since diagonal elements $\mathbf{C}_{ii} = 1$:

$$M + M(M-1)\bar{\rho} \geq 0 \tag{15}$$

Solving for $\bar{\rho}$:

$$\bar{\rho} \geq -\frac{1}{M-1} \tag{16}$$

Substituting this into the definition of $\mathcal{D}$:

$$\mathcal{D}_{\max} = 1 - \left(-\frac{1}{M-1}\right) = 1 + \frac{1}{M-1} = \frac{M}{M-1} \tag{17}$$

This upper bound is achieved when the tasks are maximally anti-correlated (simplex structure). In our experiments ($M = 7$), the theoretical maximum is $\mathcal{D} \approx 1.17$. $\qquad\square$

## M. Prompts

We set up the prompts for different datasets in our experiments following EvalScope (Team, 2024) and Verl (Sheng et al., 2024).

---

**Countdown**

User Template:

Using the numbers `{numbers}`, create an equation that equals `{target}`. You can use basic arithmetic operations (`+`, `-`, `*`, `/`) and each number can only be used once. Show your work in `<work>`...`</work>` tags. And return the final answer in `<answer>`...`</answer>` tags, for example `<answer> (1 + 2) / 3 </answer>`.

**GSM8k, Math-500, OlympiadBench**

User Template:

`{question}`. Let's think step by step and output the final answer after `####`.

**MBPP**

User Template:

You are an expert Python programmer, and here is your task: `{question}` Your code should pass these tests: `{tests}`

**USPTO-50k**

User Template: You are an expert organic chemist. Your task is to classify chemical reactions into one of 10 standard reaction categories based on the transformation type. Reaction Classes: 1: Heteroatom alkylation/arylation - N, O, S attacking C (e.g., SN2, ether formation).
2: Acylation - Forming C=O bonds with N, O, S (e.g., amide, ester formation).
3: C-C bond formation - New C-C bonds (e.g., Suzuki, Heck, Grignard).
4: Heterocycle formation - Creating rings with N, O, S.
5: Protections - Adding protecting groups (Boc, Bn, TBS, etc.).
6: Deprotections - Removing protecting groups.
7: Reductions - Adding H, removing O (e.g., ketone→alcohol, nitro→amine)
8: Oxidations - Adding O, removing H (e.g., alcohol→ketone).
9: Functional group interconversion - Changing one FG to another.
10: Functional group addition - Adding new FG to molecule (e.g., halogenation)"
Classify this reaction: Reactants ¿¿ Product: `{rxn_smiles}` Analyze the key transformation and output the class number (1-10) in `<answer>X</answer>` tags.

**ROCStories**

User Template: Below are 5 sentences from a story, but they are in the wrong order. Please arrange them in the correct chronological order.
Title: `{title}`

---

Sentence A: {sentence1} Sentence B: {sentence2} Sentence C: {sentence3} Sentence D: {sentence4} Sentence E: {sentence5}
Output the correct order as comma-separated letters (e.g., B,A,D,E,C). Only output the letters, nothing else.

**GQA**

User Template:

Look at the image and answer the question.
Question: {question}
Please reason step by step, and put your final answer within boxed{}.

## N. Color thickets

Beyond language models, we observe similar effects in diffusion models. Here, one can think of "color thickets," where certain regions of parameter space preferentially generate images with specific color palettes or visual styles. These differences may not reflect fundamentally distinct generative mechanisms, yet they still produce dense clusters of high-scoring samples under color- or style-sensitive metrics.

For instance, if the evaluation rewards blue-dominant images, a region that consistently generates bluish outputs forms a "blue thicket." More generally, thickets may arise not only from differences in reasoning, but also from biases in generative tendencies such as color, texture, or style.

*Table 4.* **Experimental results on reasoning benchmarks.** We compare RandOpt against RL-based methods (PPO, GRPO), Evolution Strategies (ES), Best-of-N and test-time majority voting (TT-MV) across model scales. Results are averaged over 3 runs with standard deviation shown in gray. **Bold** indicates best performance, underlined indicates runner-up. Details of the hyperparameters for all methods are provided in Appendix I.3.

| Model | Method | Countdown (math) ↑ | GSM8k (math) ↑ | MATH-500 (math) ↑ | OlyBench (math) ↑ | MBPP (prog.) ↑ | ROCStories (writing) ↑ | USPTO (chemistry) ↑ |
|---|---|---|---|---|---|---|---|---|
| **Qwen2.5-0.5B-Inst** | Base | $0.1_{\pm 0.1}$ | $39.9_{\pm 0.0}$ | $19.8_{\pm 0.0}$ | $4.2_{\pm 0.4}$ | $30.9_{\pm 0.2}$ | $22.0_{\pm 0.0}$ | $29.9_{\pm 0.3}$ |
| | TT-MV[†] | $0.3_{\pm 0.3}$ | $41.0_{\pm 0.1}$ | $29.2_{\pm 0.5}$ | $6.7_{\pm 0.2}$ | $37.0_{\pm 0.3}$ | $19.5_{\pm 0.2}$ | $30.2_{\pm 0.2}$ |
| | Best-of-N[‡] | $5.9_{\pm 2.0}$ | $40.5_{\pm 1.4}$ | $34.5_{\pm 0.5}$ | $10.5_{\pm 0.3}$ | $34.9_{\pm 1.3}$ | $20.3_{\pm 0.3}$ | $26.8_{\pm 1.2}$ |
| | PPO | $14.8_{\pm 2.8}$ | $43.2_{\pm 1.2}$ | $33.4_{\pm 1.7}$ | $16.1_{\pm 0.0}$ | $37.8_{\pm 3.0}$ | $19.1_{\pm 0.8}$ | $31.2_{\pm 0.6}$ |
| | GRPO | $13.0_{\pm 0.0}$ | $48.4_{\pm 0.8}$ | $33.7_{\pm 0.9}$ | $6.9_{\pm 0.1}$ | $42.8_{\pm 3.7}$ | $30.9_{\pm 1.1}$ | $31.7_{\pm 0.2}$ |
| | ES | **$14.9_{\pm 1.6}$** | $42.6_{\pm 0.7}$ | $30.5_{\pm 0.7}$ | **$16.4_{\pm 0.2}$** | $45.1_{\pm 3.8}$ | $32.0_{\pm 0.8}$ | $31.6_{\pm 0.3}$ |
| | RandOpt | $8.4_{\pm 0.3}$ | $54.1_{\pm 0.8}$ | $35.3_{\pm 0.6}$ | $15.8_{\pm 0.7}$ | **$46.2_{\pm 0.4}$** | **$32.2_{\pm 0.4}$** | $32.2_{\pm 0.9}$ |
| | ES + TT-MV | $11.2_{\pm 0.8}$ | **$61.2_{\pm 0.5}$** | **$41.3_{\pm 0.5}$** | $13.5_{\pm 0.7}$ | **$48.9_{\pm 0.7}$** | $30.5_{\pm 0.9}$ | **$32.5_{\pm 0.4}$** |
| **Qwen2.5-1.5B-Inst** | Base | $6.7_{\pm 0.1}$ | $58.8_{\pm 0.2}$ | $43.2_{\pm 0.1}$ | $13.4_{\pm 0.4}$ | $62.3_{\pm 0.2}$ | $46.7_{\pm 0.1}$ | $30.2_{\pm 0.5}$ |
| | TT-MV[†] | $30.8_{\pm 0.4}$ | $69.1_{\pm 0.3}$ | $50.0_{\pm 0.4}$ | $15.9_{\pm 0.5}$ | $68.0_{\pm 0.3}$ | $43.5_{\pm 0.5}$ | $33.0_{\pm 0.4}$ |
| | Best-of-N[‡] | $19.5_{\pm 0.2}$ | $65.4_{\pm 1.9}$ | $53.0_{\pm 0.6}$ | $20.0_{\pm 0.3}$ | $69.5_{\pm 0.3}$ | $42.2_{\pm 0.8}$ | $32.7_{\pm 1.1}$ |
| | PPO | $27.0_{\pm 0.0}$ | $71.6_{\pm 0.7}$ | $55.9_{\pm 0.3}$ | $26.3_{\pm 0.1}$ | $68.5_{\pm 0.4}$ | $51.8_{\pm 0.8}$ | $31.9_{\pm 0.2}$ |
| | GRPO | $27.5_{\pm 0.7}$ | $72.1_{\pm 0.7}$ | $54.1_{\pm 0.5}$ | $18.8_{\pm 0.8}$ | **$70.2_{\pm 0.4}$** | $53.6_{\pm 1.3}$ | $31.8_{\pm 0.0}$ |
| | ES | $44.2_{\pm 0.0}$ | $71.7_{\pm 0.9}$ | $54.1_{\pm 2.8}$ | $27.2_{\pm 1.2}$ | $69.9_{\pm 0.6}$ | **$60.2_{\pm 0.6}$** | $32.0_{\pm 0.2}$ |
| | RandOpt | **$52.7_{\pm 0.5}$** | $76.4_{\pm 0.3}$ | $59.7_{\pm 0.6}$ | **$30.4_{\pm 0.7}$** | $69.6_{\pm 0.5}$ | $48.5_{\pm 0.7}$ | **$34.3_{\pm 0.5}$** |
| | ES + TT-MV | $39.7_{\pm 0.3}$ | **$80.4_{\pm 0.8}$** | **$60.7_{\pm 0.7}$** | $28.9_{\pm 0.7}$ | $70.2_{\pm 0.6}$ | $59.1_{\pm 0.2}$ | $32.2_{\pm 0.6}$ |
| **Qwen2.5-3B-Inst** | Base | $10.0_{\pm 0.1}$ | $79.8_{\pm 0.4}$ | $58.6_{\pm 0.2}$ | $24.5_{\pm 0.2}$ | $69.5_{\pm 0.3}$ | $54.7_{\pm 0.1}$ | $38.5_{\pm 0.5}$ |
| | TT-MV[†] | $12.8_{\pm 0.4}$ | $82.5_{\pm 0.2}$ | $60.8_{\pm 0.2}$ | $21.8_{\pm 0.3}$ | $74.5_{\pm 0.5}$ | $57.3_{\pm 0.4}$ | $43.2_{\pm 0.7}$ |
| | Best-of-N[‡] | $28.5_{\pm 1.2}$ | $83.3_{\pm 1.4}$ | $62.5_{\pm 0.8}$ | $28.0_{\pm 0.6}$ | $73.0_{\pm 0.9}$ | $55.0_{\pm 0.7}$ | $44.3_{\pm 1.0}$ |
| | PPO | $35.3_{\pm 0.4}$ | $83.1_{\pm 0.2}$ | $64.1_{\pm 1.1}$ | $34.4_{\pm 0.2}$ | $76.3_{\pm 1.0}$ | $49.0_{\pm 0.6}$ | $44.7_{\pm 5.8}$ |
| | GRPO | $32.6_{\pm 0.1}$ | $83.2_{\pm 0.2}$ | $64.6_{\pm 1.0}$ | $29.0_{\pm 0.0}$ | $77.0_{\pm 0.9}$ | $56.3_{\pm 4.4}$ | $49.7_{\pm 2.1}$ |
| | ES | $55.6_{\pm 0.5}$ | $85.8_{\pm 5.1}$ | $61.9_{\pm 0.3}$ | $36.4_{\pm 0.1}$ | **$77.2_{\pm 1.2}$** | **$64.5_{\pm 0.6}$** | **$52.9_{\pm 1.0}$** |
| | RandOpt | $58.4_{\pm 0.2}$ | $87.1_{\pm 0.8}$ | **$68.7_{\pm 0.7}$** | $39.2_{\pm 0.6}$ | $75.9_{\pm 0.6}$ | $56.5_{\pm 0.3}$ | $42.3_{\pm 0.3}$ |
| | ES + TT-MV | **$61.9_{\pm 0.8}$** | **$87.9_{\pm 0.9}$** | $67.7_{\pm 0.7}$ | **$39.7_{\pm 0.4}$** | $76.3_{\pm 1.1}$ | $55.0_{\pm 0.4}$ | $39.8_{\pm 0.3}$ |
| **OLMo3-7B-Inst** | Base | $64.8_{\pm 0.2}$ | $82.9_{\pm 0.4}$ | $60.6_{\pm 0.1}$ | $28.7_{\pm 0.1}$ | $65.9_{\pm 0.2}$ | $64.0_{\pm 0.3}$ | $27.2_{\pm 0.4}$ |
| | TT-MV[†] | $66.8_{\pm 0.3}$ | $81.4_{\pm 0.4}$ | $61.5_{\pm 0.5}$ | $26.5_{\pm 0.4}$ | $60.5_{\pm 0.5}$ | $63.2_{\pm 0.4}$ | $30.2_{\pm 0.5}$ |
| | Best-of-N[‡] | $67.5_{\pm 0.8}$ | $85.0_{\pm 1.2}$ | $63.0_{\pm 0.7}$ | $30.5_{\pm 0.5}$ | $64.0_{\pm 0.9}$ | $63.5_{\pm 0.6}$ | $32.0_{\pm 1.0}$ |
| | PPO | $69.0_{\pm 0.1}$ | $88.4_{\pm 0.4}$ | $63.1_{\pm 0.3}$ | $28.0_{\pm 0.2}$ | $67.7_{\pm 0.2}$ | $64.7_{\pm 1.6}$ | $40.2_{\pm 3.2}$ |
| | GRPO | $68.5_{\pm 0.7}$ | $87.0_{\pm 0.2}$ | $63.5_{\pm 0.4}$ | $27.9_{\pm 0.6}$ | $70.8_{\pm 2.2}$ | $65.8_{\pm 0.6}$ | **$51.0_{\pm 0.3}$** |
| | ES | $71.0_{\pm 0.8}$ | $87.2_{\pm 0.2}$ | $69.9_{\pm 1.0}$ | $33.1_{\pm 0.8}$ | $72.0_{\pm 0.5}$ | $65.7_{\pm 1.4}$ | $45.2_{\pm 1.0}$ |
| | RandOpt | **$85.0_{\pm 0.7}$** | $89.5_{\pm 0.2}$ | **$73.7_{\pm 0.4}$** | $35.4_{\pm 0.4}$ | **$75.1_{\pm 0.9}$** | $64.5_{\pm 0.3}$ | $43.0_{\pm 0.5}$ |
| | ES + TT-MV | $75.6_{\pm 0.7}$ | **$90.2_{\pm 0.6}$** | $62.0_{\pm 1.1}$ | **$44.7_{\pm 0.3}$** | $72.5_{\pm 0.6}$ | **$66.0_{\pm 0.8}$** | $46.3_{\pm 0.6}$ |
| **OLMo3-7B** | Base | $9.8_{\pm 0.4}$ | $78.5_{\pm 0.4}$ | $31.3_{\pm 0.3}$ | $13.9_{\pm 0.3}$ | $29.1_{\pm 0.3}$ | $24.1_{\pm 0.1}$ | $29.0_{\pm 0.3}$ |
| | TT-MV[†] | $11.5_{\pm 0.3}$ | $79.5_{\pm 0.2}$ | $36.2_{\pm 0.2}$ | $15.7_{\pm 0.2}$ | $39.1_{\pm 0.3}$ | $42.2_{\pm 0.2}$ | $35.4_{\pm 0.2}$ |
| | Best-of-N[‡] | $18.0_{\pm 0.5}$ | $82.0_{\pm 1.0}$ | $45.0_{\pm 0.8}$ | $20.5_{\pm 0.4}$ | $38.0_{\pm 1.2}$ | $50.0_{\pm 0.8}$ | $38.0_{\pm 0.9}$ |
| | PPO | $21.9_{\pm 0.7}$ | $82.8_{\pm 0.9}$ | $51.8_{\pm 0.7}$ | $22.9_{\pm 0.5}$ | $57.9_{\pm 2.0}$ | $64.8_{\pm 0.3}$ | **$49.8_{\pm 0.3}$** |
| | GRPO | $28.8_{\pm 0.1}$ | $78.2_{\pm 0.3}$ | $52.0_{\pm 0.9}$ | $6.9_{\pm 0.6}$ | $58.5_{\pm 2.8}$ | $62.2_{\pm 2.8}$ | $48.0_{\pm 0.5}$ |
| | ES | $26.0_{\pm 0.3}$ | **$89.1_{\pm 0.6}$** | $61.0_{\pm 4.9}$ | $30.2_{\pm 0.4}$ | $61.8_{\pm 2.3}$ | $64.4_{\pm 1.0}$ | $48.1_{\pm 1.5}$ |
| | RandOpt | **$30.2_{\pm 0.2}$** | $85.0_{\pm 0.3}$ | $59.3_{\pm 0.5}$ | $28.9_{\pm 0.5}$ | $40.5_{\pm 0.2}$ | $64.5_{\pm 0.3}$ | $44.3_{\pm 0.3}$ |
| | ES + TT-MV | $23.6_{\pm 0.2}$ | $86.4_{\pm 0.5}$ | **$69.0_{\pm 0.5}$** | **$43.3_{\pm 0.2}$** | $56.8_{\pm 0.6}$ | **$76.3_{\pm 0.6}$** | $38.1_{\pm 0.8}$ |
| **Llama3.1-8B-Inst** | Base | $10.8_{\pm 0.2}$ | $79.8_{\pm 0.3}$ | $47.0_{\pm 0.0}$ | $19.2_{\pm 0.3}$ | $56.4_{\pm 0.0}$ | $51.8_{\pm 0.0}$ | $19.5_{\pm 0.3}$ |
| | TT-MV[†] | $25.6_{\pm 0.3}$ | $72.2_{\pm 0.4}$ | $49.2_{\pm 0.3}$ | $24.5_{\pm 0.5}$ | $59.5_{\pm 0.4}$ | $53.2_{\pm 0.4}$ | $25.4_{\pm 0.2}$ |
| | Best-of-N[‡] | $40.0_{\pm 1.0}$ | $83.5_{\pm 1.2}$ | $52.0_{\pm 0.9}$ | $23.0_{\pm 0.6}$ | $60.0_{\pm 1.0}$ | $54.0_{\pm 0.8}$ | $28.0_{\pm 1.1}$ |
| | PPO | $9.9_{\pm 0.1}$ | $81.6_{\pm 4.7}$ | $45.5_{\pm 2.0}$ | $16.1_{\pm 0.6}$ | $55.2_{\pm 1.5}$ | $57.8_{\pm 1.3}$ | $32.9_{\pm 4.6}$ |
| | GRPO | $10.0_{\pm 0.2}$ | $80.2_{\pm 1.1}$ | $45.1_{\pm 1.4}$ | $23.7_{\pm 0.9}$ | $61.0_{\pm 1.9}$ | $62.2_{\pm 2.8}$ | $35.0_{\pm 0.7}$ |
| | ES | $60.3_{\pm 0.5}$ | $82.4_{\pm 0.7}$ | $39.1_{\pm 1.7}$ | $22.3_{\pm 0.5}$ | $64.8_{\pm 0.8}$ | **$68.1_{\pm 5.5}$** | $34.1_{\pm 2.0}$ |
| | RandOpt | **$63.6_{\pm 0.4}$** | $86.7_{\pm 0.6}$ | $59.5_{\pm 0.8}$ | **$32.1_{\pm 0.0}$** | **$65.2_{\pm 0.9}$** | $59.0_{\pm 0.7}$ | $41.0_{\pm 0.6}$ |
| | ES + TT-MV | $62.5_{\pm 0.9}$ | **$87.5_{\pm 0.6}$** | **$62.0_{\pm 0.7}$** | **$32.1_{\pm 0.5}$** | $63.1_{\pm 0.8}$ | $65.2_{\pm 0.7}$ | **$64.5_{\pm 0.7}$** |

[†]TT-MV: Test-Time Majority Vote over test samples from a single trained model with different seeds.
[‡]Best-of-N: Pass@k metric with k=50.

*Table 5.* **RandOpt on 1D signals (approximation).** Each cell corresponds to a pretraining–post-training pair. In this experiment we plot results on the same function that we fit to during post-training; this is a test of apprpoximation only, not generalization to new functions. We also compare to Xavier initialization (Glorot & Bengio, 2010) and Kaiming initialization (He et al., 2015). For each pretraining method, we pick a perturbation noise scale, $\sigma$ that is large enough to show functional variation; this value is shown underneath the pretraining method name.

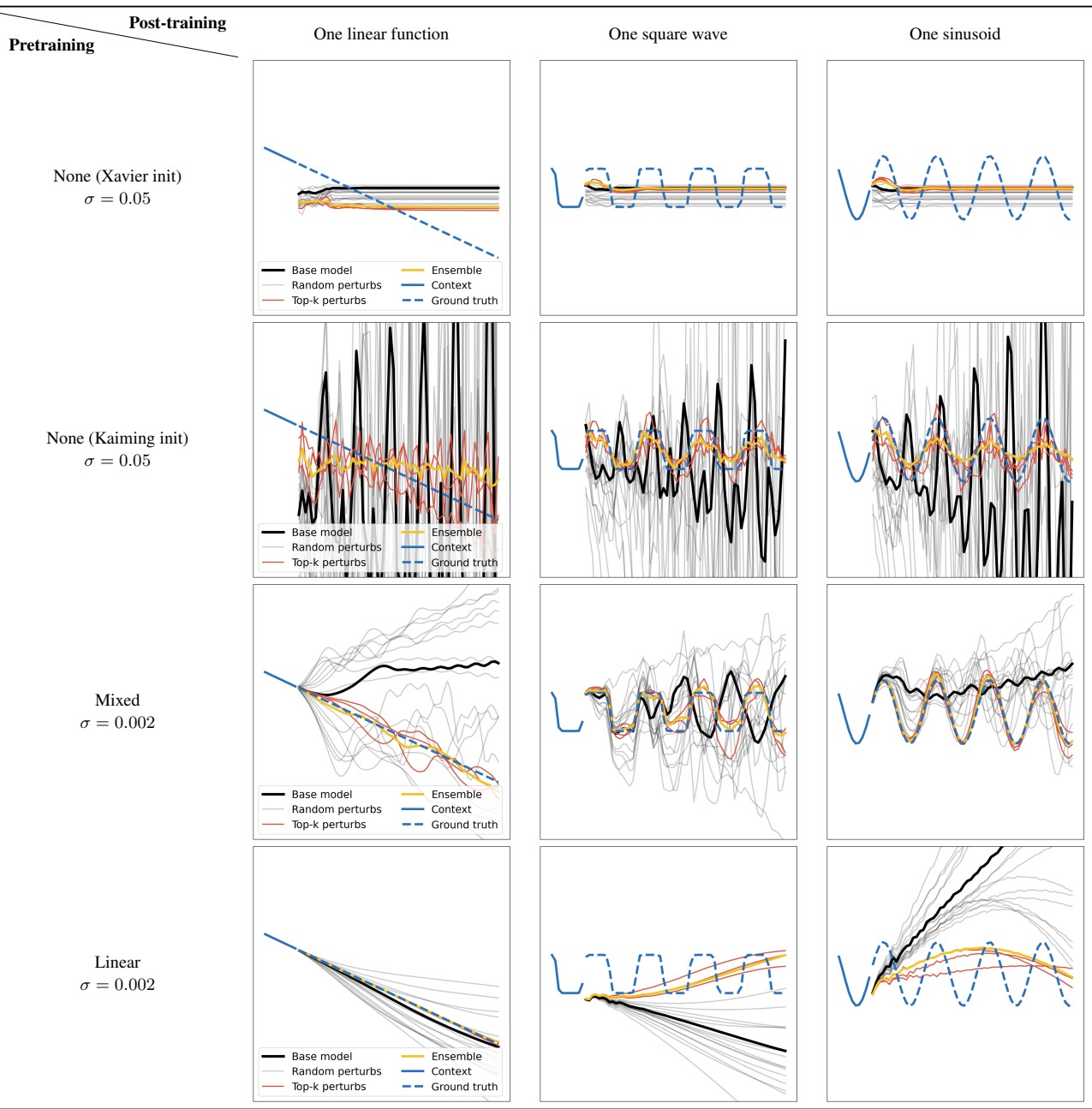

*Table 6.* **RandOpt on 1D signals (generalization).** Each row corresponds to a pretraining–post-training pair. For all rows, the test set functions are of the post-trained type, and we show three random test examples. Rightmost column shows the average mean squared error over the entire test set, for each method.

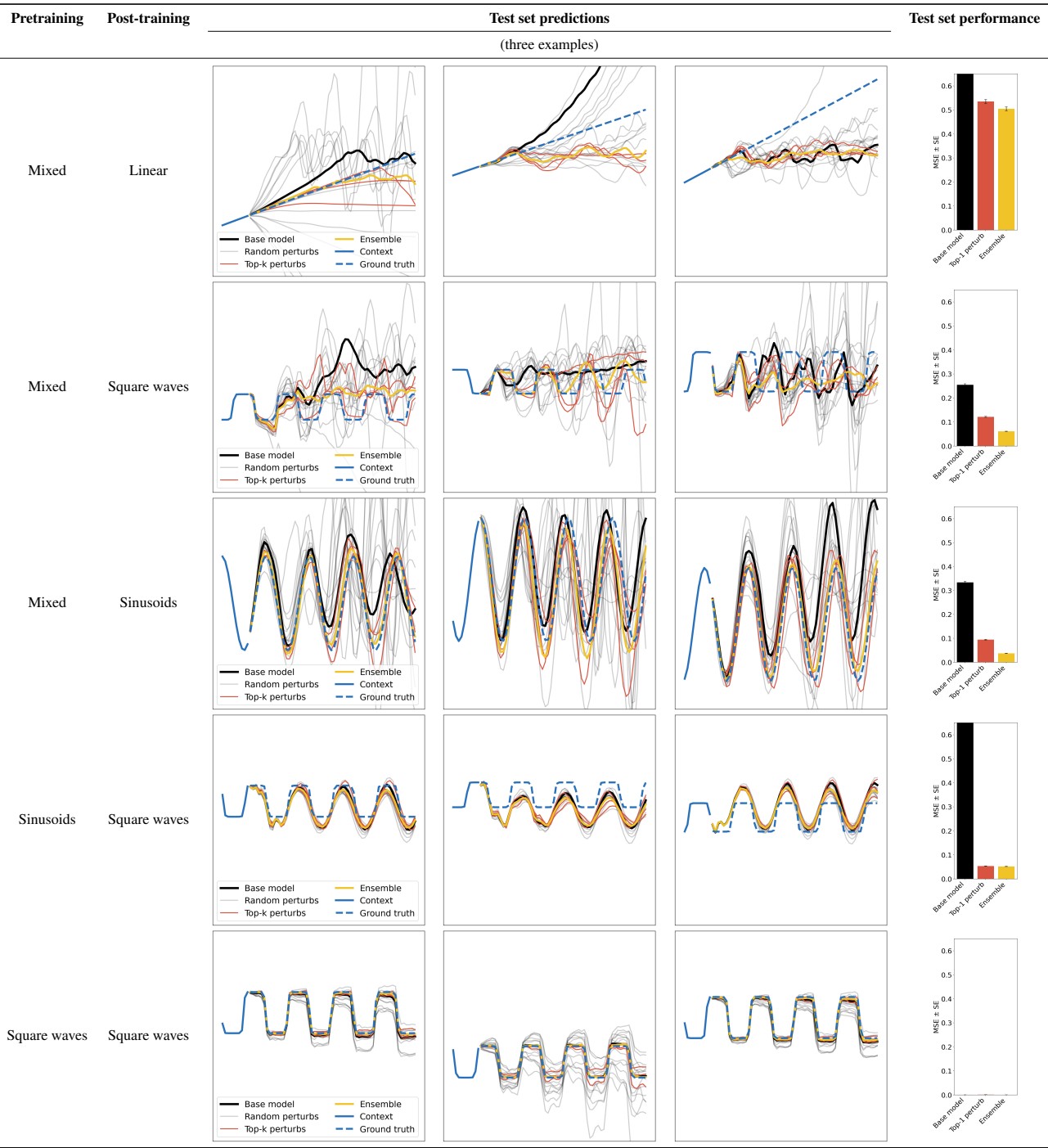

*Table 7.* **RandOpt on Text-to-Image models (Train Set).** We use the Stable Diffusion XL model (Podell et al., 2023). Images are generated from a text prompt. RandOpt selects the top-K models by scoring generated images with a target text (e.g., "blue") using GPT-5.2, and performs mean ensembling over the K models at each denoising step.

*Table 8.* **RandOpt on Text-to-Image models (Test Set).** Top-K selected on the training set and evaluated on the test set.

| | *A dog on Mars, red rocky landscape, astronaut helmet...* | *A giant octopus emerging from the ocean near...* | *A sailboat cutting through choppy ocean waves under...* | *A vintage motorcycle parked beside a brick wall...* | *A busy Tokyo subway platform during rush hour, commuters...* |
|---|---|---|---|---|---|
| **Base** | 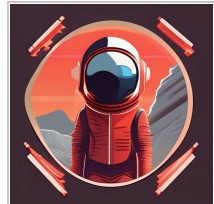 | 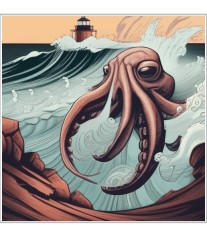 | 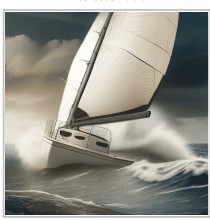 | 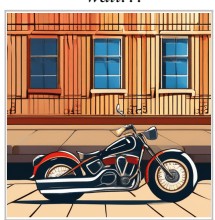 | 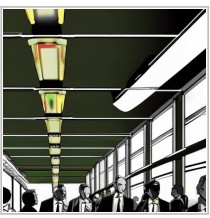 |

**Target text: Blue**

| | | | | | |
|---|---|---|---|---|---|
| **Top1** | 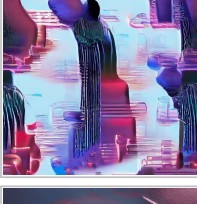 | 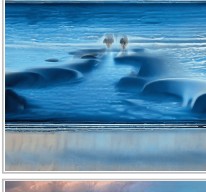 | 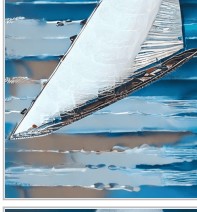 | 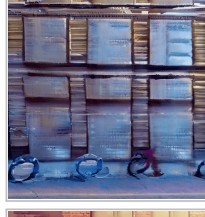 | 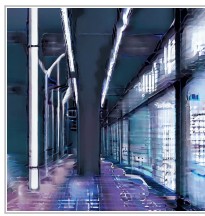 |
| **Ensemble** | 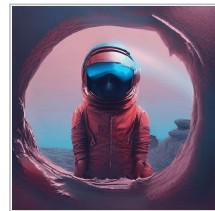 | 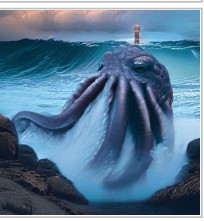 | 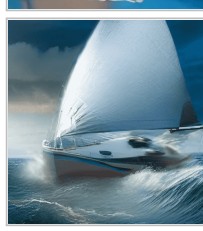 | 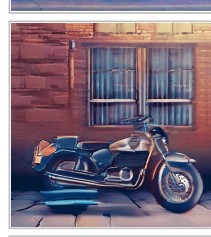 | 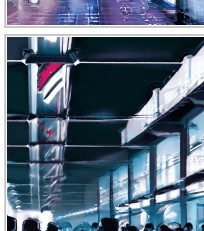 |
| **Random#1** | 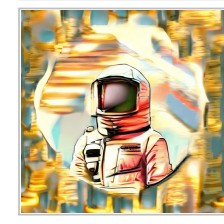 | 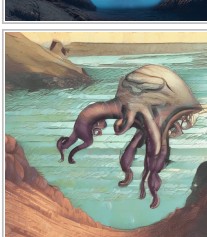 | 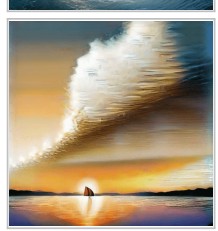 | 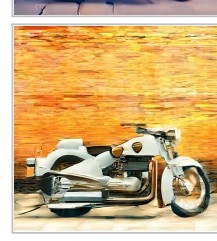 | 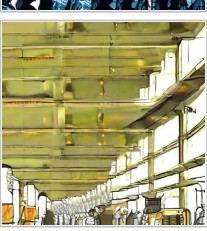 |

**Target text: Yellow**

| | | | | | |
|---|---|---|---|---|---|
| **Top1** | 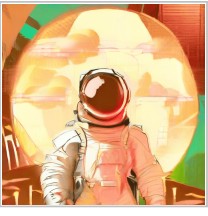 | 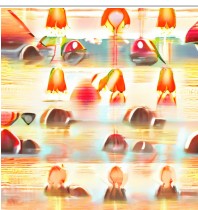 | 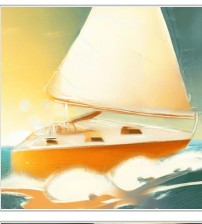 | 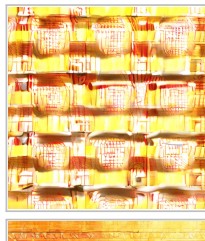 | 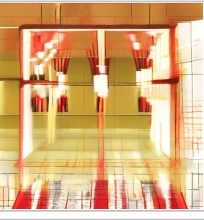 |
| **Ensemble** | 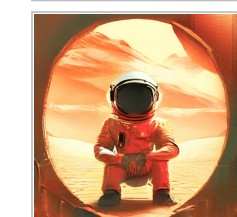 | 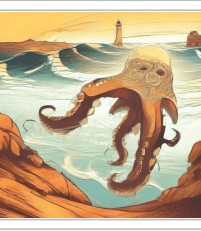 | 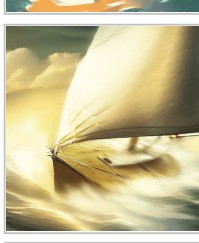 | 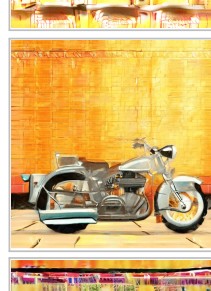 | 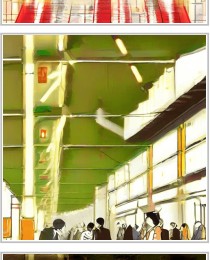 |
| **Random#2** | 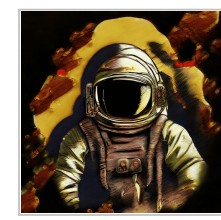 | 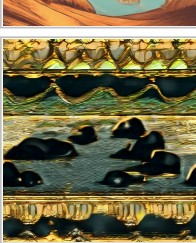 | 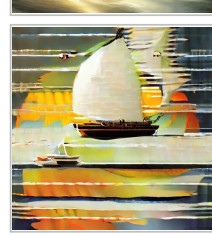 | 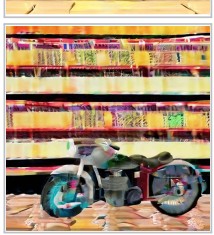 | 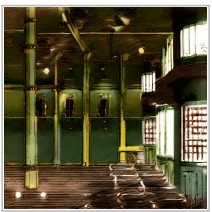 |

