# OpenReview forum: "Neural Thickets: Diverse Task Experts Are Dense Around Pretrained Weights"
_ICML.cc/2026/Conference — ICML 2026 spotlight_

### Official Review · Reviewer_2M93 · 2026-03-11

**Soundness:** 3
**Presentation:** 3
**Significance:** 4
**Originality:** 4
**Overall Recommendation:** 5
**Confidence:** 4

**Summary:**

This paper challenges the conventional iterative adaptation paradigm by proposing that large pretrained models are surrounded by "neural thickets"—dense neighborhoods of diverse, task-specific expert parameter vectors. To exploit this, the authors introduce Random Guessing & Ensembling (RGE), a gradient-free, fully parallel post-training method that samples random weight perturbations and ensembles the top candidates. The authors demonstrate that RGE is competitive with standard RLHF and ES baselines across various LLMs on reasoning and coding tasks.

**Compliance With Llm Reviewing Policy:**

Affirmed.

**Final Justification:**

The paper presents a highly original and impactful idea with strong empirical results. The rebuttal addressed my concerns well by providing additional intuition, experiments, and efficiency analysis, which strengthens my confidence in the work, so I increase my score.

**Key Questions For Authors:**

* Could authors provide a theoretical intuition or formal proof explaining the mechanism behind the monotonic scaling of solution diversity and density with model size?
* Can authors present a concrete efficiency breakdown (wall-clock GPU hours, memory footprint, I/O bottleneck analysis) for the RGE search phase compared to standard post-training methods?
* How does RGE perform when applied to visual modalities, such as adapting a pretrained Vision Transformer to a downstream dense prediction task?
* How does the performance of RGE compare directly to standard Supervised Fine-Tuning (SFT) under the exact same data constraints?

**Limitations:**

* The study is purely empirical and lacks a theoretical explanation for the formation of neural thickets.
* The approach incurs a severe $K\times$ computational penalty during inference.
* The memory and compute overhead during the parallel search phase makes the method difficult to scale on standard hardware.

**Strengths And Weaknesses:**

1. **Ambiguity in "Density" vs. "Sparsity":** While the paper provides a probabilistic definition of "Solution Density" (Equation 1), it lacks a rigorous discussion on the structural density of the weights themselves. Standard pretraining typically involves regularization that encourages weight sparsity. It is highly counter-intuitive that such optimization yields a "dense" neighborhood of solutions. The paper fails to define the specific pretraining conditions or objectives required for this thicket to emerge.
2. **Lack of Theoretical Grounding:** The paper is purely phenomenological. It heavily relies on empirical observations to claim that larger models embed more diverse experts, but lacks theoretical proof or mechanistic explanation for *why* and *how* these thickets form during pretraining. As the authors themselves admit, the fundamental motivation remains unclear.
3. **Incomplete Experimental Setup in Main Text:** Section 5.1 is overly brief and defers too many critical setup details to the appendices. The main text lacks sufficient detail regarding the evaluation pipeline, and some claims would benefit from broader citations regarding the nature of generalization.
4. **Lack of Modality Diversity:** The experiments are exclusively focused on 1D signals and Large Language Models. To convincingly claim that "Neural Thickets" are a universal property of pretrained weights, the authors must evaluate this on other modalities. For instance, testing RGE on Vision Transformers (e.g., DeiT) for downstream tasks like image classification or medical image segmentation would greatly strengthen the core hypothesis.
5. **Missing Direct Baselines:** While the paper compares RGE against RL baselines (PPO, GRPO) and a Base model, it surprisingly omits a direct ablation against standard Supervised Fine-Tuning (SFT) or traditional direct transfer learning without any parameter perturbation.
6. **Obscured Efficiency and Computational Overhead:** The paper normalizes training compute by FLOPs (Appendix E.1), but completely ignores practical efficiency metrics. While the authors explicitly acknowledge that *inference* cost increases by $K$ times, they do not report the massive wall-clock GPU time and peak memory consumption required to generate, store, and evaluate a population of up to 100,000 perturbed billion-parameter models during the RGE search phase.

---

> ### Author Rebuttal · Authors · 2026-03-31
>
> We thank Reviewer 2M93 for the detailed and helpful feedback and address each concern below.
>
> **Weakness 1 - Ambiguity in "Density" vs. "Sparsity"**
>
> This is a nice observation. The results show that good solutions can have non-sparse weights, which is counterintuitive but consistent with the fact that most LLM pretraining does not use sparsity regularizers. Most LLM pretraining uses AdamW, where the decoupled weight decay corresponds to L2 regularization, which does not encourage sparsity. [1,2]
>
> [1] Bai, et al. "Qwen technical report."
> [2] Grattafiori, et al. "The llama 3 herd of models."
>
> **Weakness 2 & Q1 When does the thicket emerge & theoretical intuition**
>
> In Section 4, we show a toy setting where pretraining on a diverse set of tasks induces this effect, and random perturbations can select the test task of interest.
>
> One theoretical intuition is that the thicket regime emerges in overparameterized models trained on diverse, partially conflicting tasks. We view post-training as sampling in a local smooth neighborhood around pretrained weights $w^*$.
>
> Under Gaussian perturbations $\delta=\sigma z$, where $z \sim \mathcal{N}(0, I)$, two factors govern how solution density scales with model size $D$:
>
> 1) Overparameterization brings w closer to good solutions for each task, reducing $\|\delta_t^*\| / \sigma$ and increasing the chance of finding good solutions.
>
> 2) Tasks have low effective dimensionality $d_{\text{eff}} \ll D$, so only a small fraction of perturbations affect task-relevant directions [3]. As $D$ grows while $d_{\text{eff}}$ stays stable, random perturbations are less likely to hurt performance.
>
> Together, small $\|\delta_t^*\| / \sigma$ makes good solutions reachable, low $d_{\text{eff}}$ makes them robust, and different tasks using different subspaces lead to diverse solutions.
>
>
> [3] Salimans, et al. "Evolution strategies as a scalable alternative to reinforcement learning."
>
> **Weakness 3 - Incomplete experimental setup in main text**
>
> Due to page limits, we moved some setup details to the Appendix. We will move the key experimental settings back to the main text for clarity.
>
> **Weakness 4 & Q3. Beyond language modality**
>
> Besides LLMs, we also conduct experiments on VLMs. We evaluate on the GQA dataset, which contains questions requiring understanding of objects, and relations in images. We perturb the model while keeping the visual encoder frozen, and run RandOpt with N = 5000 and K = 50. This improves accuracy on GQA by 12.4%.
>
> **Rebuttal Table 7** RandOpt improves Accuracy (%) on GQA.
>
> | Model              | Method  | GQA  |
> |-------------------|---------|------|
> | Qwen2.5-VL-3B-Inst | Base    | 56.6 |
> | Qwen2.5-VL-3B-Inst | RandOpt | 69.0 |
>
> **Weakness 5 & Q4 How does the performance of RandOpt compare to SFT?**
>
> We did not include the SFT baseline because it often brings limited gains and can even hurt performance [4], due to sparse rewards leading to weak or misleading supervision. Many LLM post-training papers also do not have SFT results [5,6]. Experiments on Qwen2.5-1.5B-Instruct show that RandOpt clearly outperforms SFT.
>
> **Rebuttal Table 8** Supervised fine-tuning Baseline
>
> | Model              | Method  | Countdown | GSM8k | MATH-500 | OlyBench | MBPP | ROCStories | USPTO |
> |-------------------|---------|-----------|-------|----------|----------|------|------------|-------|
> | Qwen2.5-1.5B-Inst | Base    | 6.7       | 58.8  | 43.2     | 13.4     | 62.3 | 46.7       | 30.2  |
> |                   | SFT     | 8.2       | 59.5  | 45.0     | 17.0     | 63.0 | 45.3       | 30.8  |
> |                   | RandOpt | 52.7      | 76.4  | 59.7     | 30.4     | 69.6 | 48.5       | 34.3  |
>
> [4] Jiang, et al. "A Study of Post-Training Methods for Large Language Models."
> [5] Guo, et al. "Deepseek-r1"
> [6] Hatamizadeh, et al. "Self-Feedback-Driven LLM Reasoning."
>
> **Weakness 6. Computational Overhead**
>
> The inference cost can be reduced from K$\times$ to a single model via distillation, with only a 2% increase in training cost. See our first response to Reviewer HkYd on the same point for details.
>
> **Weakness 6 & Q2. Efficiency**
>
> We address the reviewer's concern regarding efficiency and clarify that RandOpt actually has a shorter wall-clock time (O(1)) and comparable peak memory to GRPO or PPO.
>
> In a speedrun to 70% accuracy on the Countdown task, RandOpt requires only 3.2 minutes (200 GPUs, N=2000, K=50, model: OLMo3-7B-Instruct), while GRPO takes 1.1 hours and PPO takes 10 hours under standard settings. In terms of total compute, RandOpt uses $3 \times 10^{18}$ FLOPs, compared to $6 \times 10^{18}$ for GRPO and $2 \times 10^{19}$ for PPO.
>
> Regarding peak memory, RandOpt does not store multiple perturbed models simultaneously. Instead, perturbations are generated on-the-fly from random seeds, evaluated, and discarded in a streaming fashion. Therefore, the peak memory usage is approximately that of a single model forward pass, independent of the $N$.

---

> > ### Author Rebuttal · Reviewer_2M93 · 2026-04-01
> >
> > The author addressed all of my concerns and I fully recognize the contribution of this work. So I raised my score to 5.

---

### Official Review · Reviewer_cAw5 · 2026-03-11

**Soundness:** 3
**Presentation:** 1
**Significance:** 3
**Originality:** 4
**Overall Recommendation:** 5
**Confidence:** 1

**Summary:**

This paper argues that the local weight-space around pretrained models changes with scale. In small or weakly pretrained models, task-improving perturbations are rare, but in larger pretrained models the neighborhood around the pretrained weight vector becomes dense with many possible task-improving perturbations, and these perturbations are heterogeneous specialists rather than variants of a single generalist.

Motivated by that picture, the authors propose a simple post-training method, RGE (Random Guessing & Ensembling): sample many perturbations around pretrained weights,, evaluate them on a small training/validation set, select the top K, and ensemble their outputs by majority vote.

**Compliance With Llm Reviewing Policy:**

Affirmed.

**Final Justification:**

The authors addressed my concerns!

**Key Questions For Authors:**

- I found some of the results graphs difficult to interpret, but I'm a bit confused by Figure 7. I thought the main point was that RGE does better when the base model does better. Maybe this is what Figure 9 is supposed to show, but then I think the y axis should be normalized by the base model performance, right? Actually looking at the graph it seems like that trend doesn't seem to hold? Also I think there's a typo - it says RRE not RGE in fig 9.

**Limitations:**

Yes they discussed the limitations.

**Strengths And Weaknesses:**

Strengths
- This is a super cool idea. I really like the insight and the intro had an interesting connection to prior thought in the field.
- The idea of solution density seems well formulated and novel. Figure 5 is well done and seems to provide strong evidence for your two hypotheses.
- I like the implication that it becomes easier to posttrain larger, better pretrained models.
- The experiments seem sound.

Weaknesses
- This statement seems somewhat under supported. You've done one alternative strategy, but I'm not sure why it implies all of those "will do".
>  In that regime, it doesn’t matter much which method you use – gradient-based search, evolutionary algorithms, and brute-force parallel selection all will do.
- The method requires K inferences at inference time. The authors acknowledge this, but it probably makes the method infeasible to actually use.
- The graphs are really confusing to read and the main point is not always clear. Figure 4 is very hard to interpret, especially 4a.

---

> ### Author Rebuttal · Authors · 2026-03-31
>
> We thank Reviewer cAw5 for the useful suggestions and for recognizing the novelty of our idea and the soundness of our experiments. We address each concern below.
>
> **Weakness 1 - One statement seems under-supported - ‘In that regime, it doesn’t matter much which method you use – gradient-based search, evolutionary algorithms, and brute-force parallel selection all will do.’**
>
> This statement is supported by both the experimental results on seven datasets and six models (**Table 1** in the paper) and the accuracy landscape of LLMs (**Figure 2** in the paper). As shown in **Table 1**, we evaluate all methods mentioned in the statement: PPO and GRPO (gradient-based search), ES (evolutionary algorithms), and RandOpt (brute-force parallel selection). The results show that, across datasets spanning math reasoning, programming, writing, and chemistry, and across different models, all of these methods achieve good performance.
>
> **Figure 2** shows that once in the thicket regime, solutions are dense around the pretrained weights, so even random guessing has a high chance of finding good solutions, let alone commonly used state-of-the-art methods for LLM post-training such as gradient-based search and evolutionary algorithms.
>
> **Weakness 2 - K inferences at inference time**
>
> The inference cost can be reduced from K$\times$ to a single model via distillation, with only a 2% increase in training cost.
>
> **Rebuttal Table 6** Distilling the top-K RandOpt ensemble into a single model achieves performance comparable to the ensemble on GSM8K.
>
> | Model              | Method  | GSM8K |
> |-------------------|---------|-------|
> | Qwen2.5-1.5B-Inst. | Base    | 58.8  |
> |                   | Distill | 74.9  |
> |                   | RandOpt | 76.4  |
> | Qwen2.5-3B-Inst.   | Base    | 79.8  |
> |                   | Distill | 84.3  |
> |                   | RandOpt | 87.1  |
>
> We perform distillation on Qwen2.5-1.5B-Instruct and Qwen2.5-3B-Instruct (see **Rebuttal Table 6**). Using the top-50 models, we generate 25,000 responses from 500 training examples. Each sample includes the input question, a reasoning trace, and the final answer.
> For distillation, we use hard samples: for each input, we generate eight candidate answers and keep those where more than half are incorrect.
>
> **Weakness 3 - Confusion about figure4(a)**
>
> Figure 4(a) measures the performance of different perturbed models across various tasks. Therefore, the more zigzag a curve appears, the larger the performance variation of the same model across tasks, which indicates the models are specialist rather than the generalist. We have improved the figure highlighting only four representative curves, making it easier to read.
>
> **Q1 Had some difficulty interpreting Figure 7**
>
> Figure 7 presents results compared to the baseline. As shown, in the vast majority of cases, RandOpt outperforms GRPO; the only cases where it does not are when the base model is very weak.

---

> > ### Author Rebuttal · Reviewer_cAw5 · 2026-04-03
> >
> > Fully resolved!

---

### Official Review · Reviewer_HkYd · 2026-03-13

**Soundness:** 4
**Presentation:** 4
**Significance:** 3
**Originality:** 3
**Overall Recommendation:** 6
**Confidence:** 3

**Summary:**

The authors presents a novel perspective to investigate and understand the task-specific loss landscape around pretrianed weights. The core argument of this manuscript can be conceptualized as: pretrained weights reside within a dense "thickets" of diverse task experts, such that by utilizing local parameter perturbation, one can enhance the model's performance across most scenarios through ensemble methods. The authors have conducted extensive analysis to quantify the density and diversity of such local thickets with respect to the scale of model capacity. And empirically, their experiments indicate that random perturbe and ensemble effectively achieves comparable performance with mainstream  RL and evolution strategies.

**Compliance With Llm Reviewing Policy:**

Affirmed.

**Final Justification:**

The authors' rebuttal have successfully addressed my initial concerns. And they have provided an effective technical method that further reinforces the practical applicability and depth of the study. Therefore, I recommend accepting it.

**Key Questions For Authors:**

1. Have you tried applying more refined weight perturbations on smaller-scale models? After all, RGE currently seems to perform poorly on small models. Could this indicate that the entire neural thickets phenomenon fails to align with small models because applying a global Gaussian perturbation cannot capture the loss landscape surrounding small pre-trained models?
2. Besides the selection ratio, is the size of $D_{train}$ and the selection process sensitive? How is it sampled and does it affect the expert candidates?
3. (minor question) The pentagrams in Figure 1 (pretrained weights) and Figure 2 (best perturbation) are not consistent in meaning. However, the authors state in the Figure 1 caption, “see Fig 2 for a version with real data.” Therefore, I recommend that the authors unify the representation of the pentagrams across both figures to maintain consistency and improve readability.

**Limitations:**

yes.

**Strengths And Weaknesses:**

**Strength**:

1. This paper is technically sound and novel in the scope of LLM research. To better illustrate the concept of neural thickets, the authors designed targeted quantitative metrics for in-depth and comprehensive analysis and observation, making the paper's fundamental assumptions compelling.

2. Their concrete experimental evidence consist numerous common LLMs and benchmarks, along with detailed ablation and scaling experiments, lending the conclusions practical value and guidance.

3. The paper is clearly written, making it easy for readers to follow and comprehend. Although the proposed RGE method involves relatively complex computations, its underlying structure is straightforward and effective (particularly for pre-trained models) avoiding potential pitfalls associated with post-training approaches.

-----
**Weakness**:

1. As the authors have mentioned in the manuscript, the proposed methods RGE performs multiple runs of inference and therefore is
time consuming and induces much more computation. But given the current RGE is training-free and has simple algorithm logic, there should be more elegant improvements that better leverage this property of neural thickets, right?

2. The author analyzes and interprets the pre-trained parameters of the LLM as a unified whole, but lacks further in-depth analysis of the composition of finer-grained structures within the training process or the network itself. Moreover, the proposed RGE remains relatively vague regarding parameter manipulation across the entire model.

---

> ### Author Rebuttal · Authors · 2026-03-31
>
> We thank Reviewer HkYd for the thorough review and for recognizing our work’s technical soundness, novelty, and clarity. We address each point below.
>
> **Weakness 1.1 - Multiple runs of inference**
>
> The inference cost can be reduced from K$\times$ to a single model via distillation, with only a 2% increase in training cost.
>
> **Rebuttal Table 1** Distilling the top-K RandOpt ensemble into a single model achieves performance comparable to the ensemble on GSM8K.
>
> | Model              | Method  | GSM8K |
> |-------------------|---------|-------|
> | Qwen2.5-1.5B-Inst. | Base    | 58.8  |
> |                   | Distill | 74.9  |
> |                   | RandOpt | 76.4  |
> | Qwen2.5-3B-Inst.   | Base    | 79.8  |
> |                   | Distill | 84.3  |
> |                   | RandOpt | 87.1  |
>
> We perform distillation on Qwen2.5-1.5B-Instruct and Qwen2.5-3B-Instruct (see **Rebuttal Table 1**). Using the top-50 models, we generate 25,000 responses from 500 training examples. Each sample includes the input question, a reasoning trace, and the final answer.
> For distillation, we use hard samples: for each input, we generate eight candidate answers and keep those where more than half are incorrect.
>
> **Weakness 1.2 - There can be more elegant improvements that better leverage neural thickets.**
>
> We agree, RandOpt is minimal, serving as a probe for the thicket phenomenon. Several promising directions exist:
>
> Adaptive perturbation distributions. One could learn structured perturbations (e.g., low-rank or layer-specific) targeting directions of highest solution density.
> Hybrid approaches. Combining RandOpt's parallel sampling with a few local refinement steps.
> Smarter ensembling. Weighted ensembling based on per-sample confidence, or MoE-style routing that selects different experts for different inputs.
>
> **Weakness 2 - Lack of finer-grained structural analysis (layer-wise, etc.)**
>
> We fine-tune all parameters because we want to avoid introducing additional tricks, such as selectively “better” layers or parameters.
> As shown in the **Rebuttal Table 2**, we conducted layer-wise perturbation experiments on OLMo3-7B-Instruct (GSM8K, N=1000, K=50). Even perturbing only the last 25% of layers achieves good performance (~2% accuracy improvement over the base model).
>
> **Rebuttal Table 2** The results of using Finer-grained perturbations
>
> | Perturbation Target     | Accuracy |
> |-------------------------|----------|
> | All layers (full model) | 86.0%    |
> | Attention layers only   | 79.1%    |
> | MLP layers only         | 81.7%    |
> | Last 25% of layers      | 84.9%    |
> | First 25% of layers     | 76.2%    |
>
> **Q1. Have you tried applying more refined weight perturbations on smaller-scale models?**
>
> Thank you for the good suggestion. Yes, as shown in the **Rebuttal Table 3**, we experimented with refined perturbations on Qwen2.5-0.5B-Instruct by applying different noise scales across layers (scaled according to the mean of the weights in each layer), and this helps a lot (8.4% - > 15.5%).
>
> **Rebuttal Table 3** Accuracy on the Countdown task using Qwen2.5-0.5B-Instruct (N=5000, K=50, base model accuracy=0.1%):
>
> | Strategy                  | Accuracy |
> |--------------------------|----------|
> | Global Gaussian ($\sigma$=0.005) | 8.4%     |
> | Layer-wise adaptive $\sigma$     | 15.5%    |
>
> **Q2. Sensitivity to D_train​ size and selection process?**
>
> As shown in **Rebuttal Table 4 and 5**, the performance is not sensitive to the D_train size and different sample selection strategies. We use the first 200 samples from the training set in the paper for simplicity (See Appendix E Implementation Details).
>
> **Rebuttal Table 4** The effect of D_train size on GSM8k (Qwen2.5-3B-Instruct, N=1000, K=50)
>
> | D_train size | Test Accuracy |
> |-------------|---------------|
> | 50          | 83.2%         |
> | 100         | 81.5%         |
> | 200         | 83.0%         |
> | 500         | 83.5%         |
> | 1000        | 83.9%         |
>
> **Rebuttal Table 5** The effect of different selection strategies for D_train​ with a fixed size of 200 (task: GSM8k, model:Qwen2.5-3B-Instruct, N=1000, K=50)
>
> | Selection Method                        | Test Accuracy      |
> |----------------------------------------|--------------------|
> | First 200 samples                      | 83.0%              |
> | Last 200 samples                       | 83.5%              |
> | Random 200 samples (avg over 3 splits) | 83.1% ± 0.9%       |
>
> **Q3: Inconsistent pentagram symbols in Figures 1 and 2**
>
> Thank you for pointing this out. We will unify symbols across both figures in the revision.

---

> > ### Author Rebuttal · Reviewer_HkYd · 2026-04-04
> >
> > Thank you for your rebuttal efforts and the comprehensive responses. The additional empirical validations based on the newly introduced methods, distillation process and finer-grained perturbations, are highly effective and thoroughly address my initial concerns. Consequently, I am raising my score to 6.
> >
> > I hope that you will incorporate these new methodologies, the additional experimental results, and the insightful discussions held with all reviewers into the revised manuscript. Integrating these elements will make the paper significantly more complete and provide robust support for the core arguments.
> >
> > The phenomenon of *neural thickets* presented in this work is both novel and highly interesting. I am very glad to see this kind of interpretability work applied to llms. Investigating the loss landscape and dense expert distributions around pretrained weights represents a very promising direction for the rapid deployment and efficient downstream application of foundation models. I look forward to seeing these findings and `RandOpt` broadly validated and further explored by the research community. Good luck!

---

### Decision · Program_Chairs · 2026-04-30

**Decision:**

Accept (spotlight)

**Comment:**

While the confidence of the reviewers were somewhat low, there was a consensus about the novelty and impact of the proposed neural thickets hypothesis and the demonstration of dense, diverse task experts around pretrained weights. There were initial questions about the inference cost, limited baselines, theoretical intuition, etc., and the authors' rebuttal has fully addressed all major concerns of the reviewer. Therefore the AC recommends Accept for this submission.